# Heterogeneous Label Shift: Theory and Algorithm

**Chao Xu**[1]  **Xijia Tang**[1]  **Chenping Hou**[1]

## Abstract

In open-environment applications, data are often collected from heterogeneous modalities with distinct encodings, resulting in feature space heterogeneity. This heterogeneity inherently induces label shift, making cross-modal knowledge transfer particularly challenging when the source and target data exhibit simultaneous heterogeneous feature spaces and shifted label distributions. Existing studies address only partial aspects of this issue, leaving the broader problem unresolved. To bridge this gap, we introduce a new concept of Heterogeneous Label Shift (HLS), targeting this critical but underexplored challenge. We first analyze the impact of heterogeneous feature spaces and label distribution shifts on model generalization and introduce a novel error decomposition theorem. Based on these insights, we propose a bound minimization HLS framework that decouples and tackles feature heterogeneity and label shift accordingly. Extensive experiments on various benchmarks for cross-modal classification validate the effectiveness and practical relevance of the proposed approach.

## 1. Introduction

In many real-world tasks, data are collected from open and dynamic environments, and the source and target data may exhibit simultaneous heterogeneous feature spaces and shifted label distributions. For instance, as in the cross-modal classification task from images to text shown in Figure 1, the source and target domains may share the same set of classes but differ in modality. At the same time, the the heterogeneity of modalities also causes a certain degree of label distribution shift. A similar situation also occurs in cross-language (e.g., English/Spanish) text classification task (Prettenhofer & Stein, 2010), the feature spaces are

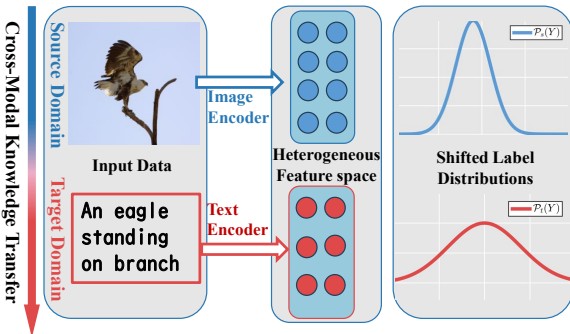

*Figure 1.* Illustration for the Heterogeneous Label Shift (HLS), in which the data is with simultaneous heterogeneous feature spaces and shifted label distributions.

referred to as vocabularies in different languages, and the features across domains are neither in the same feature space nor of the same dimensionality. Due to the differences between languages, label distribution shift is inevitable. We formulate these real-world cross-modal knowledge transfer problems as the Heterogeneous Label Shift (HLS), in which the data is with simultaneous heterogeneous feature spaces and shifted label distributions.

Reviewing traditional domain adaptation (DA) or label shift (LS) approaches, most existing works assume a homogeneous cross-domain feature space (Pan & Yang, 2010; Ben-David et al., 2010; Wang et al., 2023). The heterogeneous feature spaces block the path of existing methods to directly deal with the HLS problem. To relax this assumption, heterogeneous domain adaptation (HDA) (Dai et al., 2008; Yang et al., 2009; Day & Khoshgoftaar, 2017; Zhou et al., 2019), which tackle domain adaptation problems with different cross-domain feature spaces, has been an increased focus. In such a context, plenty of practical HDA (Zhou et al., 2014; Sukhija et al., 2016; Wang & Mahadevan, 2011; Li et al., 2019a; Shi et al., 2010; Fang et al., 2023) have been proposed. For example, Sukhija et al. proposed using shared label distributions as pivots to learn a sparse feature transformation in a supervised HDA framework (Sukhija et al., 2016). Xiao and Guo develop a semi-supervised kernel matching framework that simultaneously maps the target domain instance into the source domain instances and learns a prediction function on the labeled source instances (Xiao & Guo, 2015).

[1]College of Science, National University of Defense Technology, Changsha, 410073, China.. Correspondence to: Chenping Hou <hcpnudt@hotmail.com>.

*Proceedings of the 42nd International Conference on Machine Learning*, Vancouver, Canada. PMLR 267, 2025. Copyright 2025 by the author(s).

Despite the prominent performance achieved by these methods in applications, applying them directly to the HLS scenario will suffer from the following obstacles. (1) They ignore the adaptation to the shifted label distributions in the target domain, which is crucial for practical machine learning systems (des Combes et al., 2020; Zhao et al., 2021a; Wu et al., 2021; Bai et al., 2022; Zhou et al., 2023). (2) They inherently depend on the availability of labeled data in the target domain, which is either scarce or even absent in many tasks. (3) The mismatched label distribution inevitably disrupts the existing theoretical guarantee of HDA, so it is desirable to establish a theoretical guarantee that is simultaneous applicable to heterogeneous feature spaces and shifted label distributions. This paper aims to fill the gap in terms of both theoretical understanding and practical methods for the heterogeneous label shift setting and thereby move a step closer towards having a more complete understanding on the cross-modal knowledge transfer for heterogeneous label shift data.

To solve the aforementioned HLS problem, we first analyze the role that heterogeneous feature spaces and shifted label distributions play in the generalization ability of the model and present a novel error decomposition theorem. Motivated by this, we suggest a bound minimization HLS framework that decouples and tackles feature heterogeneity and label shift accordingly. For illustration, we devise a novel Heterogeneous Label Shift Adversarial Network (HLSAN) algorithm within the framework.

Extensive experiments on various benchmarks for cross-modal classification validate the effectiveness and practical relevance of the proposed approach. In summary, the contributions of our paper are listed as follows.

- We introduce and investigate a novel learning problem, namely, Heterogeneous Label Shift (HLS), which is rarely studied and arisen from many real application areas. To our knowledge, this may be the first attempt concerning knowledge transfer in this simultaneous heterogeneous feature spaces and shifted label distributions scenario with a theoretical guarantee.

- We present a novel error decomposition theorem that directly suggests a bound minimization HLS framework. Motivated by the theoretical analysis, we devise a Heterogeneous Label Shift Adversarial Network (HLSAN) algorithm as an illustration within the framework.

- Comprehensive experimental studies demonstrate the effectiveness of our proposal on multiple benchmarks with varying degrees of shifts for different types of cross-modal classification tasks.

## 2. Related Works

**Label shift.** Label shift (Long et al., 2018b; 2019; Cai et al., 2022a;b; Garg et al., 2020; Li et al., 2019b; Zhang et al., 2021; Zhao et al., 2021b) assumes the source and target domains have different class distributions but the same feature distribution within each class. Lipton et al. (Lipton et al., 2018b) explained that label shift is simpler than covariate shift due to the simplicity of the label space and exploited arbitrary black-box predictors to estimate the importance weights using the confusion matrix. On this basis, Azizzadenesheli (Azizzadenesheli et al., 2019b) proposed a regularized weight estimator, which obtains good statistical guarantees without a requirement on the problem-dependent minimum sample complexity, and introduced a novel regularization method to compensate for the high estimation error of the importance weights in low target sample settings (Azizzadenesheli, 2022). Alexandari et al. (Alexandari et al., 2020) combined maximum likelihood with bias-corrected calibration and introduced a principled strategy for estimating source-domain priors that improves robustness to poor calibration. Based on the above label shift methods, Garg et al. (Garg et al., 2020) proposed a unified view of label shift estimation based on confusion matrices and maximum likelihood. These approaches are not suitable for our investigations since they are limited to homogeneous domain adaptation problems and cannot handle the heterogeneous feature spaces directly.

**Heterogeneous domain adaptation.** Heterogeneous domain adaptation (HDA) (Li et al., 2014; Zhou et al., 2014; Sukhija et al., 2016; Shen & Guo, 2018; Li & Zhang, 2019; Xiao & Guo, 2015; Tsai et al., 2016; Hsieh et al., 2016; Yao et al., 2019; Fang et al., 2023) aims to enable effective cross-modal knowledge transfer between domains with inherently distinct feature spaces. This framework addresses the challenges posed by feature-space heterogeneity, ensuring meaningful alignment and adaptation across domains. Early, Duan et al.(Li et al., 2014) propose a heterogeneous feature augmentation method that transforms the data in both domains into a common subspace and then augments the projected data with original features. Later, Sukhija et al. proposed using shared label distributions as pivots to learn a sparse feature transformation in a supervised HDA framework (Sukhija et al., 2016). Xiao and Guo (Xiao & Guo, 2015) develop a semi-supervised kernel matching framework that simultaneously maps the target domain instance into the source domain instances and learns a prediction function on the labeled source instances. Shen et al. (Shen & Guo, 2018) apply the linear transformation to map source domain features onto the target domain and propose a sparse feature transformation method to tackle the unsupervised HDA problem. More recently, motivated by compatibility condition in semi-supervised probably approximately correct (PAC) theory, Fang et al. (Fang et al., 2023) propose

a generalization error estimation for target risk in semi-supervised HDA and then develop two SsHeDA algorithms based on the theoretical result. These prominent approaches can not derectly tackle HLS problem either since (1) HLS considers the unsupervised scenario, the requirement of labeled target instances remains to be a limitation for these (semi-)supervised methods. (2) These methods only focus on one aspect of our setting and they ignore that label distribution shifts.

## 3. Problem Setting and Concepts

This section outlines the problem setting and key concepts, with Table 1 providing a summary of primary notations.

### 3.1. Problem Setting

Considering a general $k$-class classification problem, let $\mathcal{Y} = \{1, \cdots, k\}$ be the label space. Denote the feature spaces of the source and target domains by $\mathcal{X}_s \subset \mathbb{R}^{d_s}$ and $\mathcal{X}_t \subset \mathbb{R}^{d_t}$, respectively. The source and target domains correspond to two distinct joint distributions, $\mathcal{P}(X_s, Y_s)$ and $\mathcal{P}(X_t, Y_t)$, where $X_s \in \mathcal{X}_s$, $X_t \in \mathcal{X}_t$, and $Y_s, Y_t \in \mathcal{Y}$ are random variables. For simplicity, we use $\mathcal{P}_s$ and $\mathcal{P}_t$ for short to denote $\mathcal{P}(X_s, Y_s)$ and $\mathcal{P}(X_t, Y_t)$, respectively. Marginal feature distributions are indicated by subscripts, for example, $\mathcal{P}_{X_s}$ represents the marginal distribution of $X_s$. The discrete label distribution is represented by specifying random variables, such as $\mathcal{P}_s(Y)$ represents the marginal distribution of $Y_s$. Then, the HLS problem is defined as follows.

*Problem* 1 (Heterogeneous Label Shift, HLS). Given sets of samples, namely a labeled source set, an unlabeled target set, and a few additional unlabeled parallel instances

$$\mathcal{S} = \{\mathbf{x}_s^i, y_s^i\}_{i=1}^{n_s} \sim \mathcal{P}_{X_s Y_s} \; i.i.d$$

$$\mathcal{T} = \{\mathbf{x}_t^i\}_{i=1}^{n_t} \sim \mathcal{P}_{X_t} \; i.i.d$$

$$\mathcal{O} = \{[\mathbf{x}_{o,s}^i, \mathbf{x}_{o,t}^i]\}_{i=1}^{n_p} \sim \mathcal{P}_{X_s} \times \mathcal{P}_{X_t} \; i.i.d \,.$$

Here $\mathcal{P}_{X_s} \times \mathcal{P}_{X_t}$ is the marginal distribution of the parallel instances and $n_p \ll n_s$, $n_p \ll n_t$. The aim of HLS is to enable effective knowledge transfer between the heterogeneous source and target domains, utilizing $\mathcal{S}$, $\mathcal{T}$, and $\mathcal{O}$ to develop a model that performs well in classifying data within the target domain.

### 3.2. Concepts

● *Feature Transformations.* Inspired by (Ben-David et al., 2010), a common strategy involves learning representations that are invariant to domain shifts. For the HLS problem, we begin by introducing the following definition.

**Definition 3.1** (Feature Transformations)**.** Given a latent

| Notation | Definition |
|---|---|
| $\mathcal{X}, \mathcal{X}_s, \mathcal{X}_t$ | latent, source, target feature space |
| $\mathcal{Y} = \{1, \cdots, k\}$ | label space |
| $X_s, X_t$ | random variables on $\mathcal{X}_s, \mathcal{X}_t$ |
| $Y_s, Y_t$ | random variables on $\mathcal{Y}$ |
| $\mathcal{P}_s, \mathcal{P}_t$ | source, target joint distributions |
| $\mathcal{P}_{X_s}, \mathcal{P}_{X_t}$ | source, target marginal distributions |
| $\mathcal{P}_s(Y), \mathcal{P}_t(Y)$ | discrete label distrbutions |
| $\mathbf{T}_s, \mathbf{T}_t$ | source, target feature transformation |
| $\mathcal{F}_s, \mathcal{F}_t$ | feature transformation space |
| $\mathcal{S}$ | The labeled source data |
| $\mathcal{T}$ | The unlabeled target data |
| $\mathcal{O}$ | The unlabeled parallel instances |

*Table 1.* Main Notations and Corresponding Definitions

space $\mathcal{X} \subset \mathbb{R}^d$, we denote

$$\mathcal{F}_s \subset \{\mathbf{T}_s : \mathcal{X}_s \mapsto \mathcal{X}\}, \;\; \mathcal{F}_t \subset \{\mathbf{T}_t : \mathcal{X}_t \mapsto \mathcal{X}\}$$

as source and target transformation spaces, respectively. The feature transformations $\mathbf{T}_s$ and $\mathbf{T}_t$ transform the heterogeneous feature spaces into a common latent feature space and induce similar distributions on $\mathcal{P}_{X_s}$ and $\mathcal{P}_{X_t}$.

● *Importance Weights.* In the label shift setting, importance weights play a crucial role.

**Definition 3.2** (Importance Weights)**.** For discrete label distributions $\mathcal{P}_s(Y)$ and $\mathcal{P}_t(Y)$ of the source and target domains, the importance weight $\mathbf{w} \in \mathbb{R}^k$ is defined as

$$\mathbf{w}(y) := \frac{\mathcal{P}_t(Y = y)}{\mathcal{P}_s(Y = y)}, \forall y \in \mathcal{Y} = \{1, \cdots, k\}.$$

● *Hypothesis Space and Risks.* Considering a general $k$-class classification task with a hypothesis space $\mathcal{H}$ consisting of scoring functions

$$\mathbf{h} : \mathcal{X} \mapsto \mathbb{R}^{1 \times |\mathcal{Y}|} = \mathbb{R}^{1 \times k}, \;\; \mathbf{x} \mapsto [h_1(\mathbf{x}), \ldots, h_k(\mathbf{x})]$$

where $h_j(\mathbf{x})(j = 1, \ldots, k)$ indicates the confidence in the prediction of label $j$. Given $\ell : \mathbb{R}^k \times \mathbb{R}^k \mapsto \mathbb{R}_{\geq 0}$ as the symmetric loss function, the expected risks of $\mathbf{h} \in \mathcal{H}$ w.r.t. loss $\ell$ under $\mathcal{P}(\mathbf{T}_t(X_t), Y_t)$ and the weighted expected risks of $\mathbf{h} \in \mathcal{H}$ w.r.t. loss $\ell$ under $\mathcal{P}(\mathbf{T}_s(X_s), Y_s)$ are given by

$$R_t(\mathbf{h} \circ \mathbf{T}_t) = \mathbb{E} \, \ell(\mathbf{h}(\mathbf{T}_t(\mathbf{x}_t)), \phi(y_t)),$$

$$R_s(\mathbf{w}, \mathbf{h} \circ \mathbf{T}_s) = \mathbb{E} \, \mathbf{w}(y_s)\ell(h(\mathbf{T}_s(\mathbf{x}_s)), \phi(y_s)).$$

where $\phi$ maps a label to the corresponding one-hot vector. Correspondingly, the weighted empirical source risk are defined as

$$\hat{R}_s(\mathbf{w}, \mathbf{h} \circ \mathbf{T}_s) = \frac{1}{n_s} \sum_{i=1}^{n_s} \mathbf{w}(y_s)\ell(\mathbf{h}(\mathbf{T}_s(\mathbf{x}_s^i)), \phi(y_s^i)).$$

# 4. Theoretical Analysis of HLS

To dissect the HLS problem, we make a theoretical analysis of HLS and present a novel error decomposition theorem that directly suggests a bound minimization HLS framework. Due to space limitations, the detailed proofs are listed in Appendix A.

## 4.1. Error Decomposition Theorem of HLS

**Definition 4.1** (Heterogeneous Feature Alignment, HFA). Considering two feature transformations $\mathbf{T}_s$ and $\mathbf{T}_t$ transform the heterogeneous feature spaces into a common latent feature space, i.e., $X = \mathbf{T}_s(X_s) \cup \mathbf{T}_t(X_t)$. The representation $X \in \mathcal{X}$ satisfies HFA if

$$\mathcal{P}_s(X|Y = y) = \mathcal{P}_t(X|Y = y), \forall y \in \mathcal{Y}.$$

HFA extends the concept of Generalized Label Shift (GLS) introduced in (des Combes et al., 2020), addressing the more complex challenge of aligning both feature spaces and marginal feature distributions. Specifically, HFA employs dual feature transformations to induce similar latent-space distributions for $\mathcal{P}_s$ and $\mathcal{P}_t$, thereby achieving alignment of heterogeneous feature spaces.

**Definition 4.2** (Conditional Error Gap). For a joint distribution $\mathcal{P}$ and a classifier $h$, where $\hat{Y} = \mathbf{h}(X)$, the conditional error gap of $\mathbf{h}$ is defined as follows.

$$\Delta_{\mathrm{CE}}(\mathbf{h}) := \max_{y \neq y' \in \mathcal{Y}^2} |\mathcal{P}_s(\hat{Y} = y'|Y = y) - \mathcal{P}_t(\hat{Y} = y'|Y = y)|$$

Under the HFA assumption, it follows that $\Delta_{\mathrm{CE}}(\mathbf{h}) = 0$. We proceed to establish an upper bound on the error gap between the source and target domains, which also facilitates deriving a generalization bound for the target risk.

**Theorem 4.3** (Error Decomposition Theorem). *Considering a general $k$-class classification task. Let $\mathcal{H}$ be the family of hypothesis set, and denote the hypothesis returned by the model trained with the available data $\mathcal{S}$, $\mathcal{T}$, and $\mathcal{O}$ as $\mathbf{h}$. Suppose the loss function $\ell$ is $M$-bounded and $L$-Lipschitz w.r.t. Euclidean norm. Then, for any $\delta > 0$, with probability at least $1 - \delta$ over the source domain sample of size $n_s$ and target domain sample of size $n_t$, the following inequalities holds for any $\mathbf{h} \in \mathcal{H}$,*

$$|R_t(\mathbf{h} \circ \mathbf{T}_t) - \hat{R}_s(\hat{\mathbf{w}}, \mathbf{h} \circ \mathbf{T}_s)| \leq \Delta_{CE}(\mathbf{h}) + \|\hat{\mathbf{w}} - \mathbf{w}\|_2$$

$$+ 2L\mathfrak{R}_{n_s}(\mathcal{H} \circ \mathbf{T}_s) + M\sqrt{\frac{\log(1/\delta)}{2n_s}}. \tag{1}$$

*Here, $\mathfrak{R}_{n_s}(\mathcal{H} \circ \mathbf{T}_s)$ denotes the Rademacher complexity of the hypothesis class $\mathcal{H}$ associated with $\mathbf{T}_s$. $\mathbf{w}$ and $\hat{\mathbf{w}}$ denote the true importance weights and the estimated importance weights, respectively.*

Theorem 4.3 presents a decomposition of the error gap between source and target domains, yielding a generalization bound for the target risk $R_t(\mathbf{h}, \mathbf{T}_s, \mathbf{T}_t)$. This bound consists of three key components: 1) $\Delta_{\mathrm{CE}}(\mathbf{h})$, which measures the divergence between the conditional distributions $\mathcal{P}(\hat{Y}|Y)$; 2) $\|\hat{\mathbf{w}} - \mathbf{w}\|_2$, representing the weight estimation error as the discrepancy between estimated and true importance weights; and 3) Finite sample errors. Unlike previous works [(Ben-David et al., 2010), Theorem 2; (Zhao et al., 2019), Theorem 4.1], which decompose the error gap based on the distance between marginal feature distributions ($\mathcal{P}_{X_s}, \mathcal{P}_{X_t}$) and optimal labeling functions ($f_s^X, f_t^X$), Theorem 4.3 introduces a novel decomposition that circumvents reliance on unknown optimal labeling functions, providing a distinctive and practical perspective.

Notably, the conditional error gap $\Delta_{\mathrm{CE}}(h)$ can be minimized by aligning the transformed conditional feature distributions across domains. The importance weights $\hat{\mathbf{w}}$ can be estimated with reference to the investigation of (Lipton et al., 2018a), which the authors proved to be effective for large enough sample sizes, and the distance $\|\hat{\mathbf{w}} - \mathbf{w}\|_2$ can be bounded effectively. Synthesizing conditional error gap and weight estimation error, the result suggests that, to minimize the error gap, it suffices to align the transformed conditional distributions $\mathcal{P}(X|Y = y)$ while simultaneously minimizing the weight estimation error. Before applying a bound-minimization algorithm inspired by Theorem 4.3, two key challenges must be addressed, i.e., *Heterogeneous Feature Alignment* and *Importance Weight Estimation*.

## 4.2. Heterogeneous Feature Alignment

In HLS, the absence of target domain labels prevents direct alignment of conditional label distributions. By leveraging relative class weights between domains can provide a necessary condition for HFA, we can bypass the explicit alignment of conditional feature distributions.

**Lemma 4.4** (Necessary condition for HFA). *Considering two feature transformations $\mathbf{T}_s$ and $\mathbf{T}_t$, which transform the heterogeneous source and target feature spaces into a common latent feature space, i.e., $X = \mathbf{T}_s(X_s) \cup \mathbf{T}_t(X_t)$. If $X \in \mathcal{X}$ satisfies HFA, then*

$$\mathcal{P}_t(X) = \sum_{y \in \mathcal{Y}} \mathbf{w}(y)\mathcal{P}_s(X, Y = y) =: \mathcal{P}_s^{\mathbf{w}}(X).$$

In contrast to prior approaches that align $\mathcal{P}_t(X)$ with $\mathcal{P}_s(X)$ maximum mean discrepancy (MMD) (Long et al., 2015) or aim to align $\mathcal{P}_t(\hat{Y} \otimes X)$ with $\mathcal{P}_s(\hat{Y} \otimes X)$ (Long et al., 2018a), Lemma 4.4 proposes aligning $\mathcal{P}_t(X)$ with a reweighted marginal distribution, $\mathcal{P}_s^{\mathbf{w}}(X)$.

**Theorem 4.5.** *Given the feature transformations $\mathbf{T}_s$ and $\mathbf{T}_t$, the common latent feature space $X = \mathbf{T}_s(X_s) \cup \mathbf{T}_t(X_t)$. Let $\rho := \min_{y \in \mathcal{Y}} \mathcal{P}_t(Y = y)$ and $\mathbf{w}_M :=$*

$\min_{y \in \mathcal{Y}} \mathbf{w}(y)$. *For a hypothesis* $\mathbf{h}$, *the following inequality holds.*

$$\max_{y \in \mathcal{Y}} d_{TV}(\mathcal{P}_s(X|Y=y), \mathcal{P}_t(X|Y=y))$$

$$\leq \frac{\mathbf{w}_M R_s(\mathbf{h} \circ \mathbf{T}_s) + R_t(\mathbf{h} \circ \mathbf{T}_t) + \sqrt{8d_{JS}(\mathcal{P}_s^{\mathbf{w}}(X) \parallel \mathcal{P}_t(X))}}{\rho}, \tag{2}$$

*where* $d_{TV}$ *and* $d_{JS}$ *denote the the total variation and Jensen-Shannon distance, respectively.*

Theorem 4.5 confirms that using $\mathrm{d}_{JS}$ to match $\mathcal{P}_t(X)$ and $\mathcal{P}_s(X)$ is an appropriate objective for approximating the discrepancy of feature distribution conditioned on the label. It demonstrates that if marginal feature distributions are aligned and the source error is zero, successful domain adaptation (i.e., zero target error) implies that HFA holds.

### 4.3. Importance Weight Estimation

Accompanied by feature transformations, $\mathbf{w}$ can be estimated within a latent invariant space. Drawing inspiration from the moment-matching technique proposed by Lipton et al. (Lipton et al., 2018a) for estimating $\mathbf{w}$ under label shift, we develop a method to estimate $\mathbf{w}$ under HLS by solving a regularized quadratic program (RQP). The approach begins with the following definition.

**Definition 4.6.** Let $\hat{Y} = \mathbf{h}(X)$ be the predictions. Denote $\mathbf{C} \in \mathbb{R}^{|\mathcal{Y}| \times |\mathcal{Y}|}$ as the confusion matrix of the classifier on the source domain and $\boldsymbol{\mu} \in \mathbb{R}^{|\mathcal{Y}|}$ as the distribution of predictions on the target domain, $\forall y, y' \in \mathcal{Y}$

$$\mathbf{C}_{y,y'} := \mathcal{P}_s(\hat{Y}=y, Y=y'), \quad \boldsymbol{\mu}(y) := \mathcal{P}_t(\hat{Y}=y).$$

**Lemma 4.7.** *If the Feature Transformations* $\mathbf{T}_s$ *and* $\mathbf{T}_t$ *achieved HFA, and if the confusion matrix* $\mathbf{C}$ *is invertible, then* $\mathbf{w} = \mathbf{C}^{-1}\boldsymbol{\mu}$.

Lemma 4.7 indicates that estimating the importance weight $\mathbf{w}$ under HLS does not require target domain labels. However, matrix inversion can be numerically unstable, particularly with finite sample estimates $\hat{\mathbf{C}}$ and $\hat{\boldsymbol{\mu}}$ for $\mathbf{C}$ and $\boldsymbol{\mu}$. To address this, we propose a regularized quadratic program (RQP) to estimate $\hat{\mathbf{w}}$

$$\hat{\mathbf{w}} = \arg \min_{\mathbf{w}} ||\hat{\boldsymbol{\mu}} - \hat{\mathbf{C}}\mathbf{w}||_2^2 + \lambda ||\mathbf{w} - \mathbf{w}_0||_2^2$$
$$\text{s.t. } \mathbf{w} \geq 0, \mathbf{w}^\top \mathcal{P}_s(Y) = 1. \tag{3}$$

Here, $\mathbf{w}_0$ is an initial weight incorporating prior information, and $\lambda$ is a parameter balancing the estimated loss with the regularization term. In contrast to the estimation method in (des Combes et al., 2020), the regularization term mitigates extreme label shift, making the estimated weight $\hat{\mathbf{w}}$ more robust to estimation biases of $\hat{\mathbf{C}}$ and $\hat{\boldsymbol{\mu}}$.

**Theorem 4.8.** *Denote* $\hat{\mathbf{w}}$ *as the estimated importance weight obtained by solving the RQP in Eq.(3). Then, with*

*probability at least* $1 - \delta$, *the following inequality holds*

$$||\hat{\mathbf{w}} - \mathbf{w}||_2 \leq \varepsilon(n_s, n_t, ||\boldsymbol{\theta}||_2, \delta), \tag{4}$$

*where*

$$||\hat{\mathbf{w}} - \mathbf{w}||_2 \leq c_1 \sqrt{\frac{72k}{n_s} \log\left(\frac{12k}{\delta}\right)} + c_2 \sqrt{\frac{9k}{n_t} \log\left(\frac{6k}{\delta}\right)}. \tag{5}$$

*and*

$$c_1 = \frac{(||\boldsymbol{\theta}||_2 + 1)}{\sigma_{\min}(\mathbf{C})}, \quad c_2 = \frac{1}{\sigma_{\min}(\mathbf{C})}. \tag{6}$$

$||\boldsymbol{\theta}||_2 = ||\mathbf{w} - \mathbf{w}_0||_2$ *characterizes the distance between the true weight and the initialized weight.*

Theorem 4.8 establishes the effectiveness of estimating the importance weight $\mathbf{w}$ via RQP, demonstrating that the weight estimation error is tightly bounded. This bound is directly affected by the sample sizes $n_s$ and $n_t$, as well as the number of classes $k$. Specifically, larger sample sizes $n_s$ or $n_t$ result in a tighter bound, whereas an increase in the number of classes $k$ leads to a looser bound. Moreover, the initial weight $\mathbf{w}_0$ indirectly impacts the bound; a closer alignment of $\mathbf{w}_0$ to the true weight $\mathbf{w}$ reduces $||\boldsymbol{\theta}||_2$, further tightening the bound.

## 5. Deep Neural Network Implementation

This section shows how to design theoretically guaranteed algorithm, i.e., bringing HLS theory into the reality. As discussed in above Theorems, the algorithm is carried out from the following three aspects. (1) estimate $\mathbf{w}$ via the available data, (2) align the target feature distributions with the reweighted source feature distribution and, (3) minimize the weighted souce risk. Overall, we should consider the optimization problem as follows

$$\min_{\mathbf{h} \in \mathcal{H}, \mathbf{T}_s \in \mathcal{F}_s, \mathbf{T}_t \in \mathcal{F}_t} R_s(\mathbf{w}, \mathbf{h} \circ \mathbf{T}_s) + \alpha \Delta_{\mathrm{CE}}(\mathbf{h}). \tag{7}$$

According to Lemma 4.4, we can bypasses an explicit alignment of the conditional feature distributions by aligning $\mathcal{P}_t(X)$ with a reweighted marginal distribution $\mathcal{P}_s^{\mathbf{w}}(X)$. Then Eq.(7) can be transformed into

$$\min_{\mathbf{h} \in \mathcal{H}, \mathbf{T}_s \in \mathcal{F}_s, \mathbf{T}_t \in \mathcal{F}_t} R_s(\mathbf{w}, \mathbf{h} \circ \mathbf{T}_s) + \alpha \mathrm{d}_{\mathcal{H}}(\mathcal{P}_t(X), \mathcal{P}_s^{\mathbf{w}}(X)). \tag{8}$$

To measure the distribution discrepancy we introduce the integral probability metric ((Mller, 1997), IPM).

**Definition 5.1** ($\mathcal{H}$-IPM)**.** Denote $\mathcal{H}$ as a set of real-valued functions. For two distributions $\mathcal{D}$ and $\mathcal{D}'$, the $\mathcal{H}$-IPM between distributions $\mathcal{D}$ and $\mathcal{D}'$ is

$$\mathrm{d}_{\mathcal{H}}(\mathcal{D}, \mathcal{D}') := \sup_{h \in \mathcal{H}} |\mathbb{E}_{X \sim \mathcal{D}}[h(X)] - \mathbb{E}_{X \sim \mathcal{D}'}[h(X)]|$$

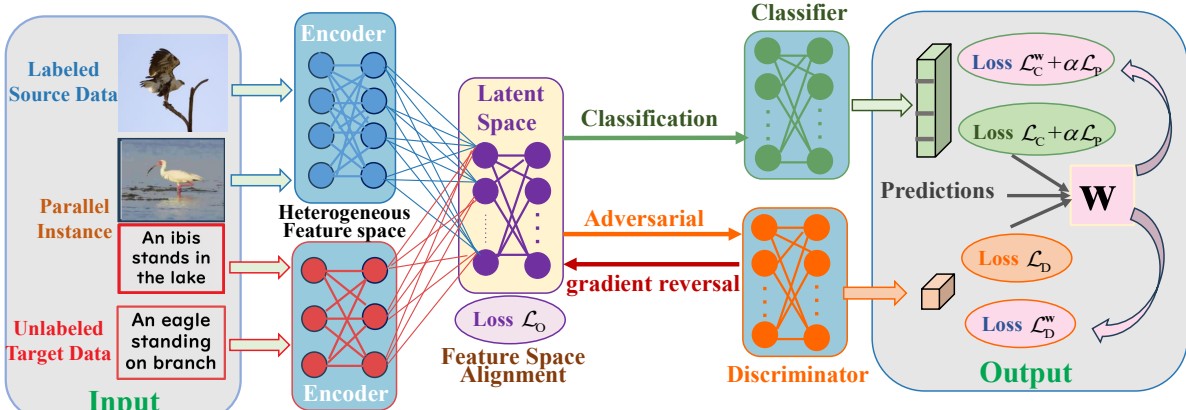

*Figure 2.* Illustration for the proposed HLSAN architecture. HLSAN consists of two main components *Feature Space Alignment* and *Feature Distribution Alignment*, which form a positive feedback loop, reinforcing each other to achieve an satisfactory result.

By approximating any function class $\mathcal{H}$ using parametrized models, e.g., neural networks, we propose a general HLS framework that is compatible with $\mathcal{H}$-integral probability metrics.

### 5.1. Network-Based HLS Algorithm

Inspired by domain-adversarial neural networks [(Ganin et al., 2016), DANN], we instantiate $\mathcal{F}_s, \mathcal{F}_t$ and $\mathcal{H}$ as fully-connected neural networks and propose a network-based algorithm *Heterogeneous Label Shift Adversarial Network* (HLSAN). As illustrated in Figure 2, the HLSAN structure consists of two main components. The first part, *Feature Space Alignment*, projects the heterogeneous space into a latent common space through two feature transformation networks, $\mathbf{T}_{s,\theta_s}$ and $\mathbf{T}_{t,\theta_t}$, parameterized by $\theta_s$ and $\theta_t$, respectively. In the second part, the distributions in the latent common space are aligned while training a classifier suited to the target domain. This second part includes two modules: a label classifier $\boldsymbol{h}_\phi$, parameterized by $\phi$, and a domain discriminator $d_\varphi$, parameterized by $\varphi$. The design of the HLSAN network is guided by two main objectives: (1) optimizing $\mathbf{T}_s, \mathbf{T}_t$, and $h$ to minimize the label classifier loss, i.e., *Weighted Source Risk Minimization*, (2) optimizing $f_s, f_t$, and $d_\varphi$ to maximize the domain discriminator loss, achieving *Feature Distribution Alignment*.

In a nutshell, *Feature Space Alignment* and *Feature Distribution Alignment* collaborate to achieve *Heterogeneous Feature Alignment*. Dynamically updated importance weights mitigate the challenges of label shift, effectively linking *Heterogeneous Feature Alignment* with *Weighted Source Risk Minimization*.

### 5.2. Loss in HLSAN

Recalling the HLS framework formulated in Eq.(8),

$$\min_{\mathbf{h}\in\mathcal{H}, \mathbf{T}_s\in\mathcal{F}_s, \mathbf{T}_t\in\mathcal{F}_t} R_s(\mathbf{w}, \mathbf{h}\circ\mathbf{T}_s) + \alpha \mathrm{d}_\mathcal{H}(\mathcal{P}_t(X), \mathcal{P}_s^{\mathbf{w}}(X)).$$

According to the investigation of (des Combes et al., 2020), $\mathrm{d}_\mathcal{H}^P(\mathcal{P}_t(X), \mathcal{P}_s^{\mathbf{w}}(X))$ is instantiated with the Jensen-Shannon Divergence. Specifically, the loss include domain discriminator loss $\mathcal{L}_{DA}^{\mathbf{w}}$ and classifier loss $\mathcal{L}_C^{\mathbf{w}}$.

• *Domain Discriminator Loss.* We align $\mathcal{P}_t(X)$ and $\mathcal{P}_s^{\mathbf{w}}(X)$ using a discriminator. For batches $\{\mathbf{x}_s^i, y_s^i\}^B$ and $\{\mathbf{x}_t^i\}^B$ of size $B$, the weighted domain discriminator loss is

$$\mathcal{L}_{DA}^{\mathbf{w}}(\{\mathbf{x}_s^i, y_s^i\}^B, \{\mathbf{x}_t^i\}^B, \mathbf{T}_s, \mathbf{T}_t, \varphi)$$
$$= -\frac{1}{B}\sum_{i=1}^B \left[\mathbf{w}(y_s^i)\log\left(d_\varphi(\mathbf{T}_s(\mathbf{x}_s^i))\right) + \log\left(1 - d_\varphi(\mathbf{T}_t(\mathbf{x}_t^i))\right)\right],$$

(9)

Moreover, we also verify that the standard ADA framework applied to $\mathcal{L}_{DA}^{\mathbf{w}}$ indeed minimizes $\mathrm{d}_{JS}(\mathcal{P}_s^{\mathbf{w}}(X) \parallel \mathcal{P}_t(X))$ in Theorem 5.2.

**Theorem 5.2.** *Let $\mathcal{D}(x,y)$ and $\mathcal{D}'(x)$ be two density distributions, and $w(y)$ be a positive function such that $\int \mathcal{D}(y)w(y)dy = 1$. Let $\mathcal{D}^w(x) = \int \mathcal{D}(x,y)w(y)dy$ denote the $w$-reweighted marginal distribution of $x$ under $\mathcal{D}$. Define the following function with respect to $d(x)$*

$$I(d) := \mathbb{E}_{(x,y)\sim\mathcal{D}, x'\sim\mathcal{D}'}\left[-w(y)\log(d(x)) - \log(1 - d(x'))\right].$$

*The analysis shows that when*

$$d(x) = d^*(x) = \frac{\mathcal{D}^w(x)}{\mathcal{D}^w(x) + \mathcal{D}'(x)},$$

*$I(d)$ attains the minimum value*

$$I(d^*(x)) = \log(4) - d_{JS}(\mathcal{D}^w(x) \parallel \mathcal{D}'(x)).$$

Instantiating $\mathcal{D}(x, y)$ and $\mathcal{D}'(x)$ to $\mathcal{P}_s^{\mathbf{w}}(X)$ and $\mathcal{P}_t(X)$ proves that the $\mathcal{L}_{DA}^{\mathbf{w}}$ leads to minimizing $d_{JS}(\mathcal{P}_s^{\mathbf{w}}(X) \parallel \mathcal{P}_t(X))$.

• *Classifier Loss.* As for *Weighted Souce Risk Minimization*, we adopt the commonly used cross-entropy loss. For batches $\{\mathbf{x}_s^i, y_s^i\}^S$ of size $S$, $\mathcal{L}_C$ is formulated as

$$\mathcal{L}_C^{\mathbf{w}}(\{\mathbf{x}_s^i, y_s^i\}^B, \{\mathbf{x}_t^i\}^B, \mathbf{T}_s, \phi)$$
$$= -\frac{1}{B} \sum_{i=1}^{B} \mathbf{w}(y_s^i) \log \left( \mathbf{h}_\phi(\mathbf{T}_s(\mathbf{x}_s^i))_{y_s^i} \right). \quad (10)$$

• *Overall Loss.* Combining $\mathcal{L}_{DA}^{\mathbf{w}}$ and $\mathcal{L}_C^{\mathbf{w}}$, we obtain the overall loss

$$\mathcal{L} = \mathcal{L}_C^{\mathbf{w}} - \alpha \mathcal{L}_{DA}^{\mathbf{w}}, \quad (11)$$

where the hyper-parameter $\lambda$ is used to tune the trade-off between these two quantities during the learning process.

Everything seemed ready, however, a potential risk in *Feature Space Alignment* is that poor initialization can lead to failed knowledge transfer, as demonstrated in (Zhao et al., 2019; Ye et al., 2021). We further illustrate this scenario with a simple example in the Appendix B.3. To mitigate this risk, we aim to find a appropriate representation for both source and target domain data within the common space. Fortunately, parallel instances provide a natural bridge for aligning source and target modalities, as they share a common label (albeit unknown). By aligning the marginal feature distributions of parallel instances across the two modalities, we can effectively prevent extreme cases of negative transfer. Specifically, we add a loss $\mathcal{L}_O$ for *Feature Space Alignment* based on the parallel instances. For batches $\{\mathbf{x}_{o,s}^i, \mathbf{x}_{o,t}^i\}^S$ of size $S$, $\mathcal{L}_O$ is formulated as

$$\mathcal{L}_O(\theta_s, \theta_t) = \sum_{i=1}^{S} \left\| \mathbf{T}_{s,\theta_s}(\mathbf{x}_{o,s}^i) - \mathbf{T}_{t,\theta_t}(\mathbf{x}_{o,t}^i) \right\|_2. \quad (12)$$

The parallel instances create a cross-modal channel, enabling a solution to the HLS problem. Through this channel, *Feature Space Alignment* and *Feature Distribution Alignment* form a positive feedback loop, reinforcing each other to achieve an satisfactory result.

Within this cross-modal channel, leveraging the implicit label information in parallel instances allows us to further enhance *Feature Distribution Alignment*. Specifically, $\mathrm{d}_{\mathcal{H}}^P(\mathcal{P}_t(X), \mathcal{P}_s(X))$ is instantiated with the projected maximum mean discrepancy (MMD) (Gretton et al., 2012) on the parallel instances. For batches $\{\mathbf{x}_{o,s}^i, \mathbf{x}_{o,t}^i\}^S$ of size $S$, $\mathcal{L}_P$ is formulated as

$$\mathcal{L}_P(\theta_s, \theta_t, \phi) = \frac{1}{S} \sum_{i=1}^{S} \left\| \boldsymbol{h}_\phi(\mathbf{T}_{s,\theta_s}(\mathbf{x}_{o,s}^i)) - \boldsymbol{h}_\phi(\mathbf{T}_{t,\theta_t}(\mathbf{x}_{o,t}^i)) \right\|_2.$$
$$(13)$$

Take all the losses of *Feature Distribution Alignment* into account, we obtain the overall knowledge transfer (KT) loss as

$$\mathcal{L}_{KT}(\hat{\theta}_s, \hat{\theta}_t, \hat{\phi}, \hat{\varphi}) = \mathcal{L}_C^{\mathbf{w}} + \alpha \mathcal{L}_P - \beta \mathcal{L}_{DA}^{\mathbf{w}}, \quad (14)$$

where the hyper-parameter $\alpha$ and $\beta$ are used to tune the trade-off between these three quantities during the learning process. In summary, optimizing HLSAN involves two key components, i.e, *Feature Space Alignment* and *Feature Distribution Alignment*, governed by the respective loss functions $\mathcal{L}_O$ and $\mathcal{L}_{KT}$. HLSAN alternately optimizes $\mathcal{L}_O$ and $\mathcal{L}_{KT}$ either until a convergence criterion is satisfied or for a predefined number of iterations. The more optimization and implementation details are provided in Appendix B.1 and B.2.

# 6. Experiment

In this section, we evaluate the performance of the HLS approach with other closely related methods. Subsequently, we present an in-depth analysis of the proposed CMAN method from different aspects, including ablation study, parameter sensitivity analysis and convergence behavior. Before going into the details, let us introduce datasets and baselines first.

## 6.1. Configuration

**Dataset.** We design cross-modal knowledge transfer tasks using two real-world datasets: the Multilingual Reuters Collection (Li et al., 2014) and Wikipedia (Fang et al., 2023). Detailed descriptions of these tasks are provided in Appendix B.4. To simulate shifted label distributions in the benchmark datasets, we employ the Dirichlet shift approach proposed in (Guo et al., 2020). In this setup, the target dataset is assumed to follow a uniform label distribution, achieved through sampling, while the source domain label distribution is altered. Specifically, the source label distribution is sampled from a Dirichlet distribution parameterized by the concentration factor $\gamma$. Experiments are conducted under three label distribution shift settings, $\gamma = 2$, $\gamma = 5$, and $\gamma = 10$. Additional implementation details are provided in Appendix B.2.

**Compared Methods.** We evaluate the performance of the HLSAN algorithm by comparing it against two groups of baseline methods with distinct objectives: modified label shift (LS) approaches and heterogeneous domain adaptation (HDA) methods. (1) To illustrate the challenges posed by heterogeneous spaces and show the effectiveness of the *Heterogeneous Feature Alignment* (HFA) module, we compare HLSAN with traditional LS approaches, such as BBSL (Lipton et al., 2018a) and RLLS (Azizzadenesheli et al., 2019a). These methods are tailored to adapting heterogeneous source and target domains by employing dimension-

| Method | $\gamma = 2$ | | | | $\gamma = 5$ | | | | $\gamma = 10$ | | | |
|---|---|---|---|---|---|---|---|---|---|---|---|---|
| | SP-EN | SP-FR | SP-GE | SP-IT | SP-EN | SP-FR | SP-GE | SP-IT | SP-EN | SP-FR | SP-GE | SP-IT |
| BBSL | 23.7±1.4 | 18.7±0.9 | 7.7±0.8 | 15.6±1.0 | 13.8±1.6 | 18.7±1.3 | 9.2±1.1 | 20.3±1.2 | 15.2±1.3 | 18.9±1.7 | 15.8±1.3 | 7.0±1.2 |
| RLLS | 22.4±1.3 | 18.2±1.0 | 6.4±0.8 | 15.6±1.6 | 13.6±1.1 | 19.3±1.5 | 9.8±1.0 | 21.0±1.1 | 14.3±1.2 | 18.7±1.4 | 15.4±1.1 | 6.9±1.1 |
| TNT | 26.7±5.8 | 25.2±3.5 | 18.9±2.5 | 22.4±3.7 | 25.2±4.4 | 24.0±4.0 | 20.2±2.0 | 19.4±2.4 | 25.0±3.1 | 20.4±3.7 | 21.9±2.6 | 24.7±2.3 |
| JEMA | 48.6±3.1 | 45.7±3.4 | 37.2±3.8 | 32.5±2.9 | 46.5±4.5 | 42.5±2.0 | 41.1±3.7 | 39.9±4.0 | 45.1±4.4 | 40.2±3.1 | 40.9±5.1 | 39.3±2.9 |
| HLSAN | **53.5±3.1** | **54.7±2.0** | **50.2±3.1** | **46.4±3.3** | **57.4±2.0** | **53.7±1.6** | **53.5±2.0** | **50.8±2.9** | **59.9±2.3** | **57.3±4.4** | **56.3±1.8** | **55.1±1.2** |

*Table 2.* Accuracy (%) with standard error on Multilingual Reuters Collection dataset from *Text→Text*. The best accuracy among all algorithms are highlighted in boldface.

| Method | $\gamma = 2$ | $\gamma = 5$ | $\gamma = 10$ |
|---|---|---|---|
| | Wiki T-I | | |
| BBSL | 16.11±2.99 | 12.94±4.72 | 14.56±3.35 |
| RLLS | 18.26±3.88 | 11.07±4.15 | 9.02±4.88 |
| TNT | 60.26±4.24 | 54.07±6.38 | 56.25±5.54 |
| JEMA | 72.02±5.01 | 68.71±3.05 | 82.28±2.13 |
| HLSAN | **81.73±3.20** | **77.61±2.52** | **85.83±0.65** |

*Table 3.* Accuracy (%) with standard error on Wikipedia dataset from *Image→Text*. The best accuracy among all algorithms are highlighted in boldface.

ality reduction techniques to enforce an equal number of features across the two domains. (2) To evaluate the effectiveness of HLS in eliminating the dependency on labeled target domain data and addressing label shift, we conduct comparisons with HDA methods, including TNT (Chen et al., 2016) and JMEA (Fang et al., 2023). For a fair comparison, we provide only a minimal amount of labeled target domain data (one sample per class) to the SsHDA methods, closely approximating the unsupervised setting.

## 6.2. Experiments Result

Based on the aforementioned datasets, we encounter two types of cross-modal knowledge transfer tasks, i.e., *Text→Text* and *Text→Image*. Table 2 and 3 report the comparison results and we can obtain the following conclusions. 1) HLSAN consistently outperforms the comparison baseline methods. 2) The modified LS performs the worst, highlighting that heterogeneous feature spaces introduce new challenges for traditional LS methods. 3) The performance of SsHDA methods heavily depends on label availability in the target domain and significantly declines under inadequate supervision.

## 6.3. Insight Analyses

### 6.3.1. ABLATION STUDY

We perform ablation experiments on four tasks, SP → EN, SP → FR, SP → GE and SP → IT, under Dirichlet shift ($\gamma = 10$) to evaluate the contributions of HLSAN components. HLSAN is divided into three variants. (1) $w/o$ **P**, excluding the parallel instances loss $\mathcal{L}_P$ (13); (2) $w/o$ **W**, excluding the importance weight **w** from (14); and (3) $w/o$ **D**, excluding the domain discriminator loss $\mathcal{L}_{DA}^{\mathbf{w}}$ (9). Results, shown in Figure 3, reveal two key findings. 1) Removing any component degrades performance, demonstrating the importance of each term. 2) Incorporating with importance weight enhances performance, indicating the necessity of aligning label distribution shifts.

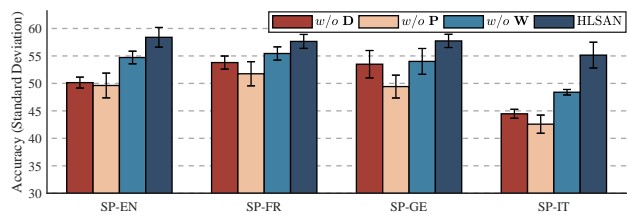

*Figure 3.* The accuracy (%) of ablation study of HLSAN on four cross-modal knowledge transfer tasks with $\gamma = 10$.

### 6.3.2. PARAMETER ANALYSIS

The HLSAN algorithm includes two key hyperparameters, $\alpha$ and $\beta$, whose impact on performance is analyzed through parameter sensitivity experiments on two tasks: SP → EN and SP → IT, under the Dirichlet shift scenario with $\gamma = 5$. The values of $\alpha$ and $\beta$ are systematically varied within the range $[0.01, 0.05, 0.1, 0.5, 1, 5]$. The results, presented in Figure 4, demonstrate that the performance of HLSAN is sensitive to these parameters, underscoring the importance of their proper tuning. Notably, HLSAN achieves prominent

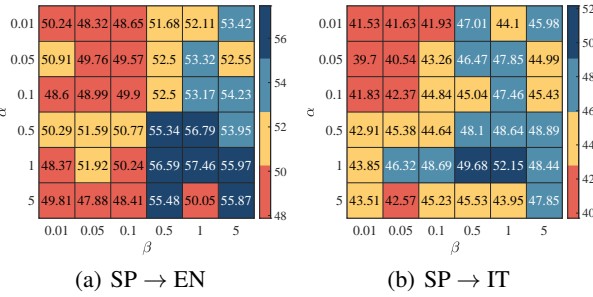

**(a) SP → EN**  **(b) SP → IT**

*Figure 4.* The sensitivity analysis of parameters $\alpha$ and $\beta$ for HLSAN on two cross-modal knowledge transfer tasks with $\gamma = 5$.

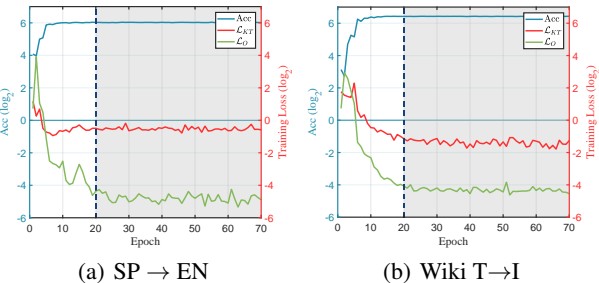

**(a) SP → EN**  **(b) Wiki T→I**

*Figure 5.* The convergence behavior of HLSAN on two cross-modal knowledge transfer tasks with $\gamma = 5$.

performance when $\alpha$ and $\beta$ are within the range $\{0.5, 1\}$ for both two tasks, indicating that the parameter interval $[0.5, 1]$ is more suitable for HLSAN.

### 6.3.3. CONVERGENCE BEHAVIOR

The HLSAN network utilizes alternating optimization for its two components: *Feature Space Alignment* and *Feature Distribution Alignment*. Figure 5 illustrates the evolution of the losses $\mathcal{L}_O$ and $\mathcal{L}_{KT}$ for these components over training epochs, alongside the accuracy curve to depict the knowledge transfer process. As training progresses, both losses steadily decrease and stabilize, while accuracy improves until reaching a plateau. These observations suggest that *Feature Space Alignment* and *Feature Distribution Alignment* interact synergistically, forming a positive feedback loop that enhances overall performance.

## 7. Conclusion

This paper addresses the critical yet underexplored challenge of Heterogeneous Label Shift (HLS), characterized by simultaneous feature space heterogeneity and label distribution shifts in cross-modal knowledge transfer. We bridge the gap in both theoretical understanding and practical methods for the HLS setting, advancing our comprehension of cross-modal knowledge transfer in the presence of heterogeneous

label shift data. Building on these insights, we developed a bound minimization framework to effectively decouple and address feature heterogeneity and label shift. Our findings highlight the importance of tackling complex, real-world distribution shifts and lay a strong foundation for future research in this area.

## Impact Statement

This paper presents work whose goal is to advance the field of Machine Learning. There are many potential societal consequences of our work, none which we feel must be specifically highlighted here.

## Acknowledgments

This work was partially supported by the NSF for Distinguished Young Scholars under Grant No. 62425607, the Key NSF of China under Grant No. 62136005. Chenping Hou is the corresponding author.

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

# A. Omitted Proofs

## A.1. Proof of Error Decomposition Theorem 4.3

*Proof.* The proof of Theorem 4.3 will start with addition and subtraction based on $|R_t(h, \mathbf{T}_t) - \hat{R}_s(h, \mathbf{T}_s)|$. By addition and subtraction we have

$$
R_t(\mathbf{h} \circ \mathbf{T}_t) - \hat{R}_s(\hat{\mathbf{w}}, \mathbf{h} \circ \mathbf{T}_s) = \underbrace{R_t(\mathbf{h} \circ \mathbf{T}_t) - R_s(\mathbf{w}, \mathbf{h} \circ \mathbf{T}_s)}_{(a)} + \underbrace{R_s(\mathbf{w}, \mathbf{h} \circ \mathbf{T}_s) - R_s(\hat{\mathbf{w}}, \mathbf{h} \circ \mathbf{T}_s)}_{(b)}
$$
$$
+ \underbrace{R_s(\hat{\mathbf{w}}, \mathbf{h} \circ \mathbf{T}_s) - \hat{R}_s(\hat{\mathbf{w}}, \mathbf{h} \circ \mathbf{T}_s)}_{(c)}. \tag{15}
$$

Here $(a)$ is the heterogeneous space alignment error, $(b)$ is the weight estimation error and $(c)$ is the finite sample error.

**Bounding term** $(a)$  Remember that $R_t(\mathbf{h} \circ \mathbf{T}_t)$ is the expected risks of $h \in \mathcal{H}$ w.r.t. loss $\ell$ under $\mathcal{P}_{\mathbf{T}_t(X_t)Y_t}$ and $R_s(\mathbf{w}, \mathbf{h} \circ \mathbf{T}_s)$ is the weighted expected risks of $h \in \mathcal{H}$ w.r.t. loss $\ell$ under $\mathcal{P}_{\mathbf{T}_s(X_s)Y_s}$. $\mathbf{w} \in \mathbb{R}^k$ is the importance weights of the target and source label distributions. $k = |\mathcal{Y}|$ is the cardinality of the finite domain of $\mathcal{Y}$. Let $\hat{Y} = h(X)$ be the prediction of $h$, then we have

$$
|R_t(\mathbf{h} \circ \mathbf{T}_t) - R_s(\mathbf{w}, \mathbf{h} \circ \mathbf{T}_s)| = |\mathbb{E}\, \ell(\mathbf{h}(\mathbf{T}_t(\mathbf{x}_t)), \phi(y_t)) - \mathbb{E}\, \mathbf{w}(y_s)\ell(h(\mathbf{T}_s(\mathbf{x}_s)), \phi(y_s))|
$$
$$
\overset{\textcircled{1}}{=} \left| \sum_{i \neq j} \mathcal{P}_{Y_t}(Y_t = j)\mathcal{P}_t(\hat{Y} = i | Y_t = j) - \sum_{i \neq j} \mathbf{w}(Y = j)\mathcal{P}_{Y_s}(Y_s = j)\mathcal{P}_s(\hat{Y} = i | Y_s = j) \right|
$$
$$
\leq \sum_{i \neq j} \left| \mathcal{P}_{Y_t}(Y_t = j)\mathcal{P}_t(\hat{Y} = i | Y_t = j) - \mathbf{w}(Y = j)\mathcal{P}_{Y_s}(Y_s = j)\mathcal{P}_s(\hat{Y} = i | Y_s = j) \right|
$$
$$
= \sum_{i \neq j} \mathcal{P}_{Y_t}(Y_t = j) \left| \mathcal{P}_t(\hat{Y} = i | Y_t = j) - \mathcal{P}_s(\hat{Y} = i | Y_s = j) \right|
$$
$$
\leq \sum_{i \neq j} \mathcal{P}_{Y_t}(Y_t = j) \max_{y \neq y' \in \mathcal{Y}^2} \left| \mathcal{P}_s(\hat{Y} = y' | Y = y) - \mathcal{P}_t(\hat{Y} = y' | Y = y) \right|
$$
$$
= \Delta_{\mathrm{CE}}(\mathbf{h}). \tag{16}
$$

Here, equality $\textcircled{1}$ holds due to the law of total probability.

**Bounding term** $(b)$  Recalling that $R_s(\mathbf{w}, \mathbf{h} \circ \mathbf{T}_s), R_s(\hat{\mathbf{w}}, \mathbf{h} \circ \mathbf{T}_s)$ are the weighted expected risks of $h \in \mathcal{H}$ w.r.t. loss $\ell$ under $\mathcal{P}_{\mathbf{T}_s(X_s)Y_s}$. $\mathbf{w}, \hat{\mathbf{w}} \in \mathbb{R}^k$ are the true importance weights and the estimated importance weights. $k = |\mathcal{Y}|$ is the cardinality of the finite domain of $\mathcal{Y}$. Let us define $\tilde{\ell} \in \mathbb{R}^k$ with $\tilde{\ell}_j = \mathbb{E}\, \mathbb{I}_{y^i = j} \ell(y^i; h(\mathbf{x}^i))$. Notice that by definition $\|\ell\|_1 \leq 1$ and $\|\ell\|_\infty \leq 1$ from which it follows by Hoelder's inequality that $\|\ell\|_1 \leq 1$. Therefore, for all $h$ we have via the Cauchy Schwarz inequality that

$$
|R_s(\mathbf{w}, \mathbf{h} \circ \mathbf{T}_s) - R_s(\hat{\mathbf{w}}, \mathbf{h} \circ \mathbf{T}_s)| = \left| \mathbb{E}\, \mathbf{w}(y_s^i)\ell(\mathbf{h}(\mathbf{T}_s(\mathbf{x}_s^i)), y_s^i) - \mathbb{E}\, \hat{\mathbf{w}}(y_s^i)\ell(\mathbf{h}(\mathbf{T}_s(\mathbf{x}_s^i)), y_s^i) \right|
$$
$$
= \left| (\mathbf{w}(y_s^i) - \hat{\mathbf{w}}(y_s^i))\mathbb{E}\, \ell(\mathbf{h}(\mathbf{T}_s(\mathbf{x}_s^i)), y_s^i) \right|
$$
$$
\leq \sum_{j=1}^{k} \left| (\mathbf{w}(Y = j) - \hat{\mathbf{w}}(Y = j))\tilde{\ell}_j \right| \leq \|\hat{\mathbf{w}} - \mathbf{w}\|_2 \tag{17}
$$

**Bounding term** $(c)$  Inspired by the proof of Theorem 3.1 in (Jennings & Wooldridge, 2012), we bound $(c)$ starting from the standard Rademacher complexity bound for $\mathcal{H}$. Let $\mathcal{L} \circ \mathcal{H}$ be the family of loss function associated to $\mathcal{H}$. Suppose the loss function $\ell$ is $M$-bounded and $L$-Lipschitz w.r.t. Euclidean norm. Now we defined a random variable $\Phi$ as follows,

$$
\Phi(\mathcal{S}) := \sup_{\mathbf{h} \in \mathcal{H}} R_s(\hat{\mathbf{w}}, \mathbf{h} \circ \mathbf{T}_s) - \hat{R}_s(\hat{\mathbf{w}}, \mathbf{h} \circ \mathbf{T}_s).
$$

We further define a ghost data set $\mathcal{S}'$ and the corresponding ghost loss $\hat{R}'_s(h, \hat{\mathbf{w}}, \mathbf{T}_s, \mathbf{T}_t)$, where $\mathcal{S}$ and $\mathcal{S}'$ are two samples differing by exactly one instance, and say $S_m = (\mathbf{x}_s^i, y_s^i)$ in $\mathcal{S}$ and $S'_m = (\mathbf{x}_s^i, y_s^i)'$ in $\mathcal{S}'$. Then, since the difference of

suprema does not exceed the supremum of the difference, we have

$$\Phi(\mathcal{S}) - \Phi(\mathcal{S}') \leq \sup_{\mathbf{h} \in \mathcal{H}} \hat{R}_s(\hat{\mathbf{w}}, \mathbf{h} \circ \mathbf{T}_s) - \hat{R}'_s(\hat{\mathbf{w}}, \mathbf{h} \circ \mathbf{T}_s) = \sup_{\mathbf{h} \in \mathcal{H}} \frac{\ell \circ \mathbf{h} \circ \mathbf{T}_s(S_m) - \ell \circ \mathbf{h} \circ \mathbf{T}_s(S'_m)}{n_s} \leq \frac{M}{n_s}. \qquad (18)$$

Similarly, we can obtain $\Phi(\mathcal{S}) - \Phi(\mathcal{S}') \leq M/n_s$, thus $|\Phi(\mathcal{S}) - \Phi(\mathcal{S}')| \leq M/n_s$. Then, by McDiarmids inequality, for any $\delta > 0$, with probability at least $1 - \delta/2$ over a sample $\mathcal{S}$ of size $n_s$, the following inequality holds:

$$\Phi(\mathcal{S}) \leq \mathbb{E}_{\mathcal{S}}\left[\Phi(\mathcal{S})\right] + M\sqrt{\frac{\log \frac{2}{\delta}}{2n_s}}.$$

Now we will proceed bound $\mathbb{E}_{\mathcal{S}}\left[\Phi(\mathcal{S})\right]$, the random variable $\Phi$ has the following properties

$$\mathbb{E}_{\mathcal{S}}\left[\Phi(\mathcal{S})\right] = \mathbb{E}_{\mathcal{S}}\left[\sup_{\mathbf{h} \in \mathcal{H}} R_s(\hat{\mathbf{w}}, \mathbf{h} \circ \mathbf{T}_s) - \hat{R}_s(\hat{\mathbf{w}}, \mathbf{h} \circ \mathbf{T}_s)\right] = \mathbb{E}_{\mathcal{S}}\left[\sup_{\mathbf{h} \in \mathcal{H}} \mathbb{E}\hat{R}'_s(\hat{\mathbf{w}}, \mathbf{h} \circ \mathbf{T}_s) - \hat{R}_s(\hat{\mathbf{w}}, \mathbf{h} \circ \mathbf{T}_s)\right]. \qquad (19)$$

We can rewrite as

$$\mathbb{E}_{\mathcal{S}}\left[\Phi(\mathcal{S})\right] = \mathbb{E}_{\mathcal{S}}\left[\sup_{\mathbf{h} \in \mathcal{H}} \mathbb{E}_{\mathcal{S}'}\left[\hat{R}'_s(\hat{\mathbf{w}}, \mathbf{h} \circ \mathbf{T}_s) - \hat{R}_s(\hat{\mathbf{w}}, \mathbf{h} \circ \mathbf{T}_s)\Big|\{\mathbf{x}_s^i, y_s^i\}_{i=1}^{n_s} \sim \mathcal{P}_{X_s Y_s}\right]\right] \qquad (20)$$

and swapping the sup with the expectation

$$\mathbb{E}_{\mathcal{S}}\left[\Phi(\mathcal{S})\right] \leq \mathbb{E}_{\mathcal{S}}\left[\mathbb{E}_{\mathcal{S}'}\left[\sup_{\mathbf{h} \in \mathcal{H}} \left[\hat{R}'_s(\hat{\mathbf{w}}, \mathbf{h} \circ \mathbf{T}_s) - \hat{R}_s(\hat{\mathbf{w}}, \mathbf{h} \circ \mathbf{T}_s)\Big|\{\mathbf{x}_s^i, y_s^i\}_{i=1}^{n_s} \sim \mathcal{P}_{X_s Y_s}\right]\right]\right]. \qquad (21)$$

According to the law of iterated conditional expectation, we remove the condition have expectation on both of the samples;

$$\begin{aligned}
\mathbb{E}_{\mathcal{S}}\left[\Phi(\mathcal{S})\right] &\leq \mathbb{E}_{\mathcal{S},\mathcal{S}'}\left[\sup_{\mathbf{h} \in \mathcal{H}}\left[\hat{R}'_s(\hat{\mathbf{w}}, \mathbf{h} \circ \mathbf{T}_s) - \hat{R}_s(\hat{\mathbf{w}}, \mathbf{h} \circ \mathbf{T}_s)\right]\right] \\
&= \mathbb{E}_{\mathcal{S},\mathcal{S}'}\left[\sup_{\mathbf{h} \in \mathcal{H}}\left[\frac{1}{n_s}\sum_{i=1}^{n_s} \hat{\mathbf{w}}(y_s^{i'})\ell(\mathbf{h}(\mathbf{T}_s(\mathbf{x}_s^{i'})), y_s^{i'}) - \hat{\mathbf{w}}(y_s^i)\ell(\mathbf{h}(\mathbf{T}_s(\mathbf{x}_s^i)), y_s^i)\right]\right] \\
&\stackrel{①}{=} \mathbb{E}_{\sigma,\mathcal{S},\mathcal{S}'}\left[\sup_{\mathbf{h} \in \mathcal{H}}\left[\frac{1}{n_s}\sum_{i=1}^{n_s} \sigma_i\left(\hat{\mathbf{w}}(y_s^{i'})\ell(\mathbf{h}(\mathbf{T}_s(\mathbf{x}_s^{i'})), y_s^{i'}) - \hat{\mathbf{w}}(y_s^i)\ell(\mathbf{h}(\mathbf{T}_s(\mathbf{x}_s^i)), y_s^i)\right)\right]\right] \\
&\stackrel{②}{\leq} \mathbb{E}_{\sigma,\mathcal{S}}\left[\sup_{\mathbf{h} \in \mathcal{H}}\left[\frac{1}{n_s}\sum_{i=1}^{n_s} -\sigma_i\left(\hat{\mathbf{w}}(y_s^i)\ell(\mathbf{h}(\mathbf{T}_s(\mathbf{x}_s^i)), y_s^i)\right)\right]\right] + \mathbb{E}_{\sigma\mathcal{S}'}\left[\sup_{\mathbf{h} \in \mathcal{H}}\left[\frac{1}{n_s}\sum_{i=1}^{n_s} \sigma_i\left(\hat{\mathbf{w}}(y_s^{i'})\ell(\mathbf{h}(\mathbf{T}_s(\mathbf{x}_s^{i'})), y_s^{i'})\right)\right]\right] \\
&\stackrel{③}{=} 2\mathbb{E}_{\sigma,\mathcal{S}}\left[\sup_{\mathbf{h} \in \mathcal{H}}\left[\frac{1}{n_s}\sum_{i=1}^{n_s} \sigma_i\left(\hat{\mathbf{w}}(y_s^i)\ell(\mathbf{h}(\mathbf{T}_s(\mathbf{x}_s^i)), y_s^i)\right)\right]\right] \\
&= 2\Re_{n_s}(\ell \circ \mathcal{H} \circ \mathbf{T}_s) \stackrel{④}{\leq} 2L\Re_{n_s}(\mathcal{H} \circ \mathbf{T}_s),
\end{aligned} \qquad (22)$$

where the equality ① holds by using the usual symmetrizing technique through Rademacher variables. After propagation sup, the latter inequality ② is obtained. By propagating the expectation and again symmetry in the Rademacher variable we obtain the inequality ③. Last, the inequality ④ holds due to the loss function $\ell$ is $L$-Lipschitz. At this point, we bound term $(c)$ as follows:

$$\sup_{\mathbf{h} \in \mathcal{H}}\left|R_s(\hat{\mathbf{w}}, \mathbf{h} \circ \mathbf{T}_s) - \hat{R}_s(\hat{\mathbf{w}}, \mathbf{h} \circ \mathbf{T}_s)\right| \leq 2L\Re_{n_s}(\mathcal{H} \circ \mathbf{T}_s) + M\sqrt{\frac{\log(1/\delta)}{2n_s}}, \qquad (23)$$

where $\Re_{n_s}(\mathcal{H} \circ \mathbf{T}_s)$ is Rademacher complexity of hypothesis function class $\mathcal{H}$ associated to $\mathbf{T}_s$. $\qquad \square$

## A.2. Heterogeneous Space Alignment

### A.2.1. PROOF OF LEMMA 4.4

*Proof.* According to the definition of HFA, we know that $\mathcal{P}_t(X|Y = y) = \mathcal{P}_s(X|Y = y)$. Applying the conditional probability formula, we will maintain the equality. By the definition of $\mathbf{w}$, we can obtain

$$
\begin{aligned}
\mathcal{P}_t(X) &= \sum_{y \in \mathcal{Y}} \mathcal{P}_t(Y = y) \cdot \mathcal{P}_t(X|Y = y) \\
&= \sum_{y \in \mathcal{Y}} \mathbf{w}(y) \mathcal{P}_s(Y = y) \cdot \mathcal{P}_s(X|Y = y) \\
&= \sum_{y \in \mathcal{Y}} \mathbf{w}(y) \mathcal{P}_s(X, Y = y) =: \mathcal{P}_s^{\mathbf{w}}(X).
\end{aligned}
$$

$\square$

### A.2.2. PROOF OF THEOREM 4.5

*Proof.* The proof of Theorem 4.5 essentially follow Theorem 3.4 from R.T. des Combes et al. (des Combes et al., 2020), except for Theorem 3.4 needs to be adapted to the HLS scenario.

Denote $\hat{Y} = \mathbf{h}(X)$ as the prediction for some $\mathbf{h} : \mathcal{X} \mapsto \mathcal{Y}$. Now consider any measurable subset $E \subseteq \mathcal{X}$, we would like to upper bound the following quantity:

$$
\begin{aligned}
|\mathcal{P}_s(X \in E|Y = y) - \mathcal{P}_t(X \in E|Y = y)| &= \frac{1}{\mathcal{P}_t(Y = y)} |\mathcal{P}_s(X \in E, Y = y)\mathbf{w}(y) - \mathcal{P}_t(X \in E, Y = y)| \\
&\leq \frac{1}{\rho} |\mathcal{P}_s(X \in E, Y = y)\mathbf{w}(y) - \mathcal{P}_t(X \in E, Y = y)|
\end{aligned}
\tag{24}
$$

Next, we upper bound $|\mathcal{P}_s(X \in E, Y = y)\mathbf{w}(y) - \mathcal{P}_t(X \in E, Y = y)|$. Considering the following decomposition:

$$
\begin{aligned}
\mathcal{P}_t(X \in E, Y = y) - \mathcal{P}_s(X \in E, Y = y)\mathbf{w}(y) &= \underbrace{\mathcal{P}_t(X \in E, Y = y) - \mathcal{P}_t(X \in E, \hat{Y} = y)}_{(a)} \\
&+ \underbrace{\mathcal{P}_t(X \in E, \hat{Y} = y) - \mathcal{P}_s^{\mathbf{w}}(X \in E, \hat{Y} = y)}_{(b)} + \underbrace{\mathcal{P}_s^{\mathbf{w}}(X \in E, \hat{Y} = y) - \mathcal{P}_s(X \in E, Y = y)\mathbf{w}(y)}_{(c)}.
\end{aligned}
\tag{25}
$$

And then we have

$$
|\mathcal{P}_s(X \in E, Y = y)\mathbf{w}(y) - \mathcal{P}_t(X \in E, Y = y)| \leq |(a)| + |(b)| + |(c)|
\tag{26}
$$

We bound the above three terms in turn. First, consider $\left|\mathcal{P}_t(X \in E, Y = y) - \mathcal{P}_t(X \in E, \hat{Y} = y)\right|$.

$$
\begin{aligned}
&\left|\mathcal{P}_t(X \in E, Y = y) - \mathcal{P}_t(X \in E, \hat{Y} = y)\right| \\
&= \left|\sum_{y' \in \mathcal{Y}} \mathcal{P}_t(X \in E, Y = y, \hat{Y} = y') - \sum_{y' \in \mathcal{Y}} \mathcal{P}_t(X \in E, \hat{Y} = y, Y = y')\right| \\
&\leq \sum_{y' \neq y} \left|\mathcal{P}_t(X \in E, Y = y, \hat{Y} = y') - \mathcal{P}_t(X \in E, \hat{Y} = y, Y = y')\right| \\
&\leq \sum_{y' \neq y} \left(\mathcal{P}_t(X \in E, Y = y, \hat{Y} = y') + \mathcal{P}_t(X \in E, \hat{Y} = y, Y = y')\right) \\
&\leq \sum_{y' \neq y} \left(\mathcal{P}_t(Y = y, \hat{Y} = y') + \mathcal{P}_t(\hat{Y} = y, Y = y')\right) \\
&\leq \mathcal{P}_t(\hat{Y} \neq Y) = R_t(\mathbf{h} \circ \mathbf{T}_t),
\end{aligned}
\tag{27}
$$

where the last inequality is due to the fact that the definition of error rate corresponds to the sum of all the off-diagonal elements in the confusion matrix while the sum here only corresponds to the sum of all the elements in two slices. Similarly, we can bound the third term as follows:

$$
\begin{aligned}
&\left| \mathcal{P}_s^{\mathbf{w}}(X \in E, \hat{Y} = y) - \mathcal{P}_s(X \in E, Y = y)\mathbf{w}(y) \right| \\
&= \left| \sum_{y' \in \mathcal{Y}} \mathcal{P}_t(X \in E, Y = y, \hat{Y} = y, Y = y')\mathbf{w}(y') - \sum_{y' \in \mathcal{Y}} \mathcal{P}_t(X \in E, \hat{Y} = y', Y = y)\mathbf{w}(y) \right| \\
&\leq \left| \sum_{y' \neq y} \mathcal{P}_t(X \in E, Y = y, \hat{Y} = y, Y = y')\mathbf{w}(y') - \mathcal{P}_t(X \in E, \hat{Y} = y', Y = y)\mathbf{w}(y) \right| \\
&\leq \mathbf{w}_M \sum_{y' \neq y} \Big( \mathcal{P}_t(X \in E, Y = y, \hat{Y} = y, Y = y') + \mathcal{P}_t(X \in E, \hat{Y} = y', Y = y) \Big) \\
&\leq \mathbf{w}_M \mathcal{P}_s(\hat{Y} \neq Y) \\
&\leq \mathbf{w}_M R_s(\mathbf{h} \circ \mathbf{T}_s).
\end{aligned}
\tag{28}
$$

Now we bound the last term. Recall the definition of total variation, we have

$$
\begin{aligned}
&\left| \mathcal{P}_t(X \in E, \hat{Y} = y) - \mathcal{P}_s^{\mathbf{w}}(X \in E, \hat{Y} = y) \right| \\
&= \left| \mathcal{P}_t(X \in E \wedge X \in \hat{Y}^{-1}(y)) - \mathcal{P}_s^{\mathbf{w}}(X \in E \wedge X \in \hat{Y}^{-1}(y)) \right| \\
&\leq \sup_{E' \in \mathcal{X} \text{ is measurable}} |\mathcal{P}_t(X \in E') - \mathcal{P}_s^{\mathbf{w}}(X \in E')| \\
&= \mathrm{d}_{TV}(\mathcal{P}_t(X), \mathcal{P}_s^{\mathbf{w}}(X)).
\end{aligned}
\tag{29}
$$

Combining the above three parts yields

$$
|\mathcal{P}_s(X \in E|Y = y) - \mathcal{P}_t(X \in E|Y = y)| \leq \frac{1}{\rho} \cdot \Big( \mathbf{w}_M R_s(\mathbf{h} \circ \mathbf{T}_s) + R_t(\mathbf{h} \circ \mathbf{T}_t) + \mathrm{d}_{TV}(\mathcal{P}_s^{\mathbf{w}}(X), \mathcal{P}_t(X)) \Big).
$$

Now realizing that the choice of $y \in \mathcal{Y}$ and the measurable subset $E$ is arbitrary, this leads to

$$
\max_{y \in \mathcal{Y}} \sup_E |\mathcal{P}_s(X \in E|Y = y) - \mathcal{P}_t(X \in E|Y = y)| \leq \frac{1}{\rho} \cdot \Big( \mathbf{w}_M R_s(\mathbf{h} \circ \mathbf{T}_s) + R_t(\mathbf{h} \circ \mathbf{T}_t) + \mathrm{d}_{TV}(\mathcal{P}_s^{\mathbf{w}}(X), \mathcal{P}_t(X)) \Big).
$$

From Briet and Harremoes (Briet & Harremoes, 2009), we have

$$
\mathrm{d}_{TV}(\mathcal{P}_t(X), \mathcal{P}_s^{\mathbf{w}}(X)) \leq \sqrt{8 \mathrm{d}_{JS}(\mathcal{P}_s^{\mathbf{w}}(X) \| \mathcal{P}_t(X))}.
$$

To sum up, we obtain

$$
\max_{y \in \mathcal{Y}} \mathrm{d}_{TV}(\mathcal{P}_s(X|Y = y), \mathcal{P}_t(X|Y = y)) \leqslant \frac{\mathbf{w}_M R_s(\mathbf{h} \circ \mathbf{T}_s) + R_t(\mathbf{h} \circ \mathbf{T}_t) + \sqrt{8 \mathrm{d}_{JS}(\mathcal{P}_s^{\mathbf{w}}(X) \| \mathcal{P}_t(X))}}{\rho},
$$

which completes the proof. $\qquad\square$

### A.2.3. PROOF OF THEOREM 5.2

Before proving Theorem 5.2, we first introduce the Jensen-Shannon Divergence.

**Definition A.1** (Jensen-Shannon Divergence). For two distributions $\mathcal{D}$ and $\mathcal{D}'$, the Jensen-Shannon (JS) divergence $\mathrm{d}_{JS}(\mathcal{D} \| \mathcal{D}')$ is defined as:

$$
\mathrm{d}_{JS}(\mathcal{D} \| \mathcal{D}') := \frac{1}{2}\mathrm{d}_{KL}(\mathcal{D} \| \mathcal{D}_M) + \frac{1}{2}\mathrm{d}_{KL}(\mathcal{D}' \| \mathcal{D}_M)
$$

where $\mathrm{d}_{KL}(\cdot \| \cdot)$ is the Kullback-Leibler (KL) divergence and $\mathcal{D}_M := \frac{1}{2}(\mathcal{D} + \mathcal{D}')$.

*Proof.* Recalling the function $I(d)$, we see that

$$
\begin{aligned}
I(d) &= -\int\int\int \left[w(y)\log(d(x)) + \log(1 - d(x'))\right] p(x, y)q(x')dxdx'dy \\
&= -\int \left[\int w(y)p(x, y)dy\right] \log(d(x)) + q(x)\log(1 - d(x))dx \\
&= -\int p^w(x)\log(d(x)) + q(x)\log(1 - d(x))dx.
\end{aligned}
\tag{30}
$$

From the last line, we follow the exact method from Goodfellow et al. (Goodfellow et al., 2014) to see that point-wise in $x$ the minimum is attained for $d^*(x) = \frac{p^w(x)}{p^w(x)+q(x)}$ and that $I(d^*) = \log(4) - 2\mathrm{d}_{JS}(p^w(x) \parallel q(x))$.

Instantiating $p^w(x)$ and $q(x)$ to $\mathcal{P}_s^{\mathbf{w}}(X)$ and $\mathcal{P}_t(X)$ proves that the $\mathcal{L}_{DA}^{\mathbf{w}}$ leads to minimizing $\mathrm{d}_{JS}(\mathcal{P}_s^{\mathbf{w}}(X) \parallel \mathcal{P}_t(X))$. $\quad\square$

### A.3. Importance Weight Estimation

#### A.3.1. PROOF OF LEMMA 4.7

*Proof.* According to the definition of HLS, and with the joint hypothesis $\hat{Y} = h(X)$ over both source and target domains, it is straightforward to see that the induced conditional distributions over predicted labels match between the source and target domains, i.e.:

$$
\mathcal{P}_s(\hat{Y} = \mathbf{h}(X)|Y = y) = \mathcal{P}_t(\hat{Y} = \mathbf{h}(X)|Y = y), \forall y \in \mathcal{Y}.
\tag{31}
$$

Then we can compute $\boldsymbol{\mu}(y), \forall y \in \mathcal{Y}$ as

$$
\begin{aligned}
\boldsymbol{\mu}(y) &= \mathcal{P}_t(\hat{Y} = y) \\
&= \sum_{y'\in\mathcal{Y}} \mathcal{P}_t(\hat{Y} = y|Y = y') \cdot \mathcal{P}_t(Y = y') \\
&= \sum_{y'\in\mathcal{Y}} \mathcal{P}_s(\hat{Y} = y|Y = y') \cdot \mathcal{P}_t(Y = y') \\
&= \sum_{y'\in\mathcal{Y}} \mathcal{P}_s(\hat{Y} = y, Y = y') \cdot \frac{\mathcal{P}_t(Y = y')}{\mathcal{P}_s(Y = y')} \\
&= \sum_{y'\in\mathcal{Y}} \mathbf{C}_{y,y'} \cdot \mathbf{w}(y')
\end{aligned}
\tag{32}
$$

Rewrite the above equation in matrix form yields $\boldsymbol{\mu} = \mathbf{C}\mathbf{w}$, and $\mathbf{C}$ is invertible, thus $\mathbf{w} = \mathbf{C}^{-1}\boldsymbol{\mu}$. $\quad\square$

#### A.3.2. PROOF OF THEOREM 4.8

*Proof.* The proof of Theorem 4.8 starts from Lemma 3.5 and theorem 3.7 proposed by (Pires & Szepesvári, 2012). For a squared penalized loss $\mathcal{L}^2(\boldsymbol{\theta}) + \lambda||\boldsymbol{\theta}||_2$, where $\mathcal{L}(\boldsymbol{\theta}) = ||\mathbf{b} - \mathbf{A}\boldsymbol{\theta}||_2$, $\lambda$ is the regularization parameter. Let $\hat{\boldsymbol{\theta}} = \arg\min_{\boldsymbol{\theta}}\{\mathcal{L}^2(\boldsymbol{\theta}) + \lambda||\boldsymbol{\theta}||_2\}$ with $\lambda = \Delta_{\mathbf{A}}$, $\Delta_{\mathbf{A}} = ||\hat{\mathbf{A}} - \mathbf{A}||_2$ and $\Delta_{\mathbf{b}} = ||\hat{\mathbf{b}} - \mathbf{b}||_2$, the following inequality holds with probability at least $1 - \delta$:

$$
\mathcal{L}(\hat{\boldsymbol{\theta}}) \leq \inf_{\boldsymbol{\theta}'}\{\mathcal{L}(\boldsymbol{\theta}') + 3\Delta_{\mathbf{A}}||\boldsymbol{\theta}'||_2\} + 3\Delta_{\mathbf{b}},
\tag{33}
$$

Consider a feasible $\boldsymbol{\theta} = \boldsymbol{\theta}'$ which satisfies $||\mathbf{b} - \mathbf{A}\boldsymbol{\theta}||_2 = 0$ and then we can get an upper bound on the right hand side of Eq.(33),

$$
\inf_{\boldsymbol{\theta}'}\{\mathcal{L}(\boldsymbol{\theta}') + 2\Delta_{\mathbf{A}}||\boldsymbol{\theta}'||_2\} \leq \mathcal{L}(\boldsymbol{\theta}) + 3\Delta_{\mathbf{A}}||\boldsymbol{\theta}||_2 = 3\Delta_{\mathbf{A}}||\boldsymbol{\theta}||_2.
\tag{34}
$$

Combining Eq.(33) and Eq.(34), the following conclusion is established.

$$
\mathcal{L}(\hat{\boldsymbol{\theta}}) = ||\mathbf{b} - \mathbf{A}\hat{\boldsymbol{\theta}}||_2 = ||\mathbf{A}(\boldsymbol{\theta} - \hat{\boldsymbol{\theta}})||_2 \leq 3\Delta_{\mathbf{A}}||\boldsymbol{\theta}||_2 + 3\Delta_{\mathbf{b}}.
\tag{35}
$$

Recalling the estimation of $\mathbf{w}$, i.e., $\hat{\mathbf{w}} = \arg\min_{\mathbf{w}} ||\hat{\boldsymbol{\mu}} - \hat{\mathbf{C}}\mathbf{w}||_2^2 + \lambda ||\mathbf{w} - \mathbf{w}_0||_2$. Denote $\hat{\boldsymbol{\theta}} = \mathbf{w} - \mathbf{w}_0$, $\mathbf{A} = \mathbf{C}$ and $\mathbf{b} = \boldsymbol{\mu} - \mathbf{C}\mathbf{w}_0$. We have,

$$
\begin{cases}
\Delta_{\mathbf{b}} = ||(\hat{\boldsymbol{\mu}} - \boldsymbol{\mu}) + (\mathbf{C}\mathbf{w}_0 - \hat{\mathbf{C}}\mathbf{w}_0)||_2 \leq ||\hat{\boldsymbol{\mu}} - \boldsymbol{\mu}||_2 + ||\mathbf{C} - \hat{\mathbf{C}}||_2 * ||\mathbf{w}_0||_2 \leq \Delta_{\boldsymbol{\mu}} + \Delta_{\mathbf{C}}, \\
\Delta_{\mathbf{A}} = ||(\mathbf{C} - \hat{\mathbf{C}}||_2 = \Delta_{\mathbf{C}}.
\end{cases}
\tag{36}
$$

Denote $\sigma_{\min}(\mathbf{C})$ as the minimum singular value of $\mathbf{C}$ and $||\mathbf{A}(\hat{\boldsymbol{\theta}} - \boldsymbol{\theta})||_2 = ||\mathbf{C}(\hat{\mathbf{w}} - \mathbf{w})||_2 \geq \sigma_{\min}(\mathbf{C})||\hat{\mathbf{w}} - \mathbf{w}||_2$. Thus,

$$
\begin{aligned}
||\hat{\mathbf{w}} - \mathbf{w}||_2 &\leq \frac{3}{\sigma_{\min}(\mathbf{C})} ||\mathbf{A}(\hat{\boldsymbol{\theta}} - \boldsymbol{\theta})||_2 \leq \frac{3}{\sigma_{\min}(\mathbf{C})} (\Delta_{\mathbf{A}}||\boldsymbol{\theta}||_2 + \Delta_{\mathbf{b}}) \\
&\leq \frac{3}{\sigma_{\min}(\mathbf{C})} (\Delta_{\mathbf{C}}(||\boldsymbol{\theta}||_2 + 1) + \Delta_{\boldsymbol{\mu}}).
\end{aligned}
\tag{37}
$$

Applying the Lemma 4.2 proposed by (Azizzadenesheli, 2022), with probability at least $1 - \delta$, the following inequality holds:

$$
\Delta_{\mathbf{C}} \leq 2\sqrt{\frac{2k}{n_s} \log\left(\frac{4k}{\delta}\right)}, \quad \Delta_{\boldsymbol{\mu}} \leq \sqrt{\frac{k}{n_t} \log\left(\frac{2k}{\delta}\right)}.
\tag{38}
$$

As a result, we have with probability at least $1 - \delta$ that

$$
||\hat{\mathbf{w}} - \mathbf{w}||_2 \leq \frac{(||\boldsymbol{\theta}||_2 + 1)}{\sigma_{\min}(\mathbf{C})} \sqrt{\frac{72k}{n_s} \log\left(\frac{12k}{\delta}\right)} + \frac{1}{\sigma_{\min}(\mathbf{C})} \sqrt{\frac{9k}{n_t} \log\left(\frac{6k}{\delta}\right)}.
\tag{39}
$$

Here, $||\boldsymbol{\theta}||_2 = ||\mathbf{w} - \mathbf{w}_0||_2$ characterizes the distance between the true weight and the initialized weight. $\qquad\square$

## B. Experimentation Details

### B.1. The Optimization of HLSAN

As mentioned in the main text, HLSAN alternately optimizes $\mathcal{L}_O$ and $\mathcal{L}_{KT}$ either until a convergence criterion is satisfied or for a predefined number of iterations. Regarding to $\mathcal{L}_{KT}$, the three losses contained in $\mathcal{L}_{KT}$ can be optimized simultaneously simultaneously via stochastic gradient descent (Li et al., 2017) in a unified neural network architecture introduced by HLSAN. Specifically, the optimization of $\mathcal{L}_{KT}$ is equivalent to finding a saddle point $(\hat{\theta}_s, \hat{\theta}_t, \hat{\phi}, \hat{\varphi})$ such that

$$
(\hat{\theta}_s, \hat{\theta}_t, \hat{\phi}) = \arg\min_{\hat{\theta}_s, \hat{\theta}_t, \hat{\phi}} \mathcal{L}_{KT}(\hat{\theta}_s, \hat{\theta}_t, \hat{\phi}, \hat{\varphi}),
$$

$$
\hat{\varphi} = \arg\min_{\hat{\varphi}} \mathcal{L}_{KT}(\hat{\theta}_s, \hat{\theta}_t, \hat{\phi}, \hat{\varphi}).
$$

And the saddle point can be identified as a stationary point through the following gradient updates.

$$
\theta_s \leftarrow \theta_s - \mu \left( \frac{\partial \mathcal{L}_C^{\mathbf{w}}}{\partial \theta_s} + \alpha \frac{\partial \mathcal{L}_P}{\partial \theta_s} - \beta \frac{\partial \mathcal{L}_{DA}^{\mathbf{w}}}{\partial \theta_s} \right),
$$

$$
\theta_t \leftarrow \theta_t - \mu \left( \frac{\partial \mathcal{L}_C^{\mathbf{w}}}{\partial \theta_t} + \alpha \frac{\partial \mathcal{L}_P}{\partial \theta_t} - \beta \frac{\partial \mathcal{L}_{DA}^{\mathbf{w}}}{\partial \theta_t} \right),
$$

$$
\phi \leftarrow \phi - \mu \left( \frac{\partial \mathcal{L}_C^{\mathbf{w}}}{\partial \phi} + \alpha \frac{\partial \mathcal{L}_P}{\partial \phi} \right),
$$

$$
\varphi \leftarrow \varphi - \mu\beta \frac{\partial \mathcal{L}_{DA}^{\mathbf{w}}}{\partial \varphi}.
$$

In addition, the HLSAN algorithm includes two stages, i.e., warming-up and training. In the warming-up stage, a suitable initial weight $\mathbf{w}_0$ is provided for HLSAN, and in the starting-up stage, the distribution shifts alignment of both features and labels are performed based on the initial weights. To summarize, the HLSAN algorithm is summarized in Algorithm 1.

---

**Algorithm 1** HLSAN Algorithm

---

1: **Input**: Data $\mathcal{S}$, $\mathcal{T}$ and $\mathcal{O}$; Parameters $\alpha$, $\beta$, learning rate $\mu$, warming up epochs $E_1$ and traning epochs $E_2$ and batches per epoch $B$;

2: **Output**: neural network $\{\mathbf{T}_s, \mathbf{T}_t, \mathbf{h}_\phi, d_\varphi\}$ and the predicted target labels $\hat{\mathbf{Y}}_t = \mathbf{h} \circ \mathbf{T}_t(\mathbf{X}_t)$.

3: **Initial** $f_s$, $f_t$, $\boldsymbol{h}$ and $d_\varphi$;

4: **Shuffle** datasets $\mathcal{S}$, $\mathcal{T}$ and $\mathcal{O}$;

**Warming-up**

5:    **for** $i = 1$ to $E_1$ **do**

6:      Sample batches $\{\mathbf{x}_s^i, y_s^i\}^S$ $\{\mathbf{x}_t^i\}^S$ and $\{\mathbf{x}_{o,s}^i, \mathbf{x}_{o,t}^i\}^S$;

7:      Minimize $\mathcal{L}_O$ w.r.t. $\mathbf{T}_s$ and $\mathbf{T}_t$;

8:      Maximize $\mathcal{L}_{DA}$ w.r.t. $\theta_s$ and $\theta_t$, minimize $\mathcal{L}_{DA}$ w.r.t. $\varphi$ and minimize $\mathcal{L}_C + \alpha\mathcal{L}_P$ w.r.t. $\phi$, $\theta_s$ and $\theta_t$;

9: Calculate the initial weight $\mathbf{w}_0$ according to Lemma 4.7.

**Starting-up**

10: **while** stopping criterion is not met **do**

11:    **for** $i = E_1$ to $E_1 + E_2$ **do**

12:      Sample batches $\{\mathbf{x}_s^i, y_s^i\}^S$ $\{\mathbf{x}_t^i\}^S$ and $\{\mathbf{x}_{o,s}^i, \mathbf{x}_{o,t}^i\}^S$;

13:      Minimize $\mathcal{L}_O$ w.r.t. $f_s$ and $f_t$;

14:      Maximize $\mathcal{L}_{DA}^{\mathbf{w}}$ w.r.t. $\theta_s$ and $\theta_t$, minimize $\mathcal{L}_{DA}^{\mathbf{w}}$ w.r.t. $\varphi$ and minimize $\mathcal{L}_C^{\mathbf{w}} + \alpha\mathcal{L}_P$ w.r.t. $\phi$, $\theta_s$ and $\theta_t$;

15:      Update the weight $\mathbf{w}$ by solving RQP(3).

16:    **End for**

17: **End while**

---

## B.2. Implementation Details

*Network Architecture.* In CMAN, the feature transformation network $\mathbf{T}_{s,\theta_s}$ and $\mathbf{T}_{t,\theta_t}$ are instanced with four-layer fully-connected neural networks. The label classifier $\boldsymbol{h}_\phi$ and domain discriminator $d_\varphi$ are instanced with three-layer fully-connected neural networks. All network parameters are optimized using SGD with momentum. And the learning rate is set to 0.02, The activation function is RELU.

*Parameters Setting.* There are two parameters $\alpha$ and $\beta$ in HLSAN, the value ranges of $\alpha$ and $\beta$ are set to $[0.01, 0.05, 0.1, 0.5, 1, 5]$. As for the comparison methods, we utilize the suggested default parameter settings provided by their original authors. In addition, the number of parallel instances is set to 100, i.e., $n_p = 100$. The warming-up epochs $E_1 = 20$ and the starting-up epochs $E_2 = 50$. For the Dirichlet shift, we draw $\mathcal{P}_s(Y)$ from a dirichlet distribution with concentration parameter as 10.

*Remark* B.1. Note that in the *Importance Weight Estimation*, an appropriate initial weight $\mathbf{w}_0$ should be provided before solving the RQP in Eq.(3). Specifically, we first warm up HLSAN by training without weighting for the first 20 epochs, and then we build estimators $\hat{\mathbf{C}}$ and $\hat{\boldsymbol{\mu}}$ of $\mathbf{C}$ and $\boldsymbol{\mu}$ by averaging the predictions of the classifier on the source (per true class) and target (overall) to calculate $\mathbf{w}_0$. Subsequently, the importance weight $\mathbf{w}$ is estimated on the fly during training.

## B.3. Examples of knowledge transfer failures

In the main text we mentioned that the initialization of common space in the extreme case could lead to the failure of knowledge transfer. Take a simple example, as shown in Figure 6(a), the transformed source and target domains have perfectly aligned marginal feature distributions in the latent common space, but this is an impossible knowledge transfer task, which has been fully demonstrated in (Zhao et al., 2019; Ye et al., 2021). In addioition, as highlighted in the main text, parallel instances serve as a natural bridge for aligning source and target modalities due to their shared (albeit unknown) labels. Aligning the marginal feature distributions of parallel instances across modalities effectively mitigates the risk of extreme negative transfer. To validate the effectiveness of this technique, we conducted comparative experiments between HLSAN and its variant $w/o$ **O**, which excludes the *Feature Space Alignment* parallel instances loss $\mathcal{L}_O$ (12). The results, presented in Figure 6(b), demonstrate that omitting $\mathcal{L}_O$ induces significant negative shifts, ultimately leading to the failure of the knowledge transfer task.

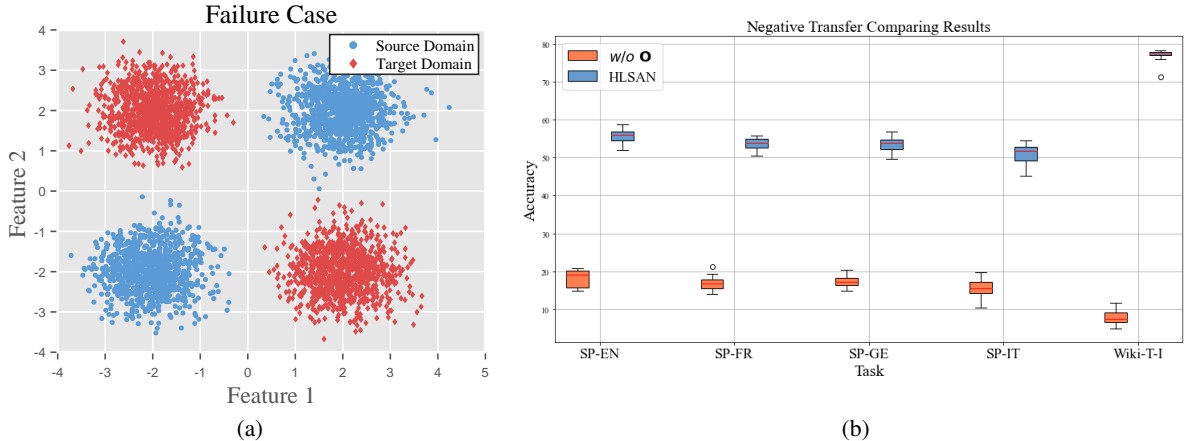

Figure 6. Examples of knowledge transfer failures. The left is a failure case, where the marginal feature distributions are perfectly aligned, but knowledge transfer fails. The right is the comparative experiment results of HLSAN and $w/o$ **O**.

### B.4. Description of the cross-modal knowledge transfer tasks

Three real-world datasets are used in the experiments, i.e., Multilingual Reuters Collection, NUS+ImageNet, and Wikipedia, which arise from different cross-modal knowledge transfer tasks.

- *Wikipedia*[1] is sourced from Wikipedia feature articles and contains 2,866 image-text pairs across 10 semantic categories. Following the the settings in (Fang et al., 2023), image features (I) are extracted using the Big Transfer-M (BiT-M) model with ResNet-101 (Kolesnikov et al., 2020), while text features (T) are obtained using the Big Bird model (Zaheer et al., 2020), which is well-suited for long sequences like Wikipedia texts. In the *Text→Image* task, text features (T) serve as the source domain, and image features (I) as the target domain.

- *Multilingual Reuters Collection*[2] contains over 11,000 news articles spanning six categories across five languages, i.e., English, French, German, Italian, and Spanish. Each text data is represented by a bag-of-words weighted by TF-IDF. Since the original data is not tractable due to its high dimensionality, following (Li et al., 2014), we perform dimensionality reduction of features using PCA with 60% energy preserved. After reduction, the feature dimensions for English, French, German, Italian, and Spanish are 1,131, 1,230, 1,471, 1,041, and 807, respectively. Inspired by the setup from (Hsieh et al., 2016; Fang et al., 2023), we designate Spanish as the source domain, while the other four languages serve as target domains, resulting in four *Text→Text* tasks.

---

[1]http://www.svcl.ucsd.edu/projects/crossmodal/

[2]http://multilingreuters.iit.nrc.ca/ReutersMultiLingualMultiView.htm

