# OpenReview forum: "Heterogeneous Label Shift: Theory and Algorithm"
_ICML.cc/2025/Conference — ICML 2025 poster_

### Official Review · Reviewer_kepV · 2025-03-05

**Overall Recommendation:** 3

**Summary:**

This paper introduces Heterogeneous Label Shift (HLS), a novel challenge in cross-modal knowledge transfer where both feature spaces and label distributions differ. It presents a new error decomposition theorem and a bound minimization framework to separately tackle feature heterogeneity and label shift. The authors propose HLSAN, an adversarial deep learning algorithm that aligns feature spaces, estimates label shifts, and improves target domain classification. Experiments show HLSAN outperforms existing methods, validating its theoretical and practical effectiveness.

**Claims And Evidence:**

Most of the claims and evidence makes sense by themselves. However, my major concern is on the interrelationship between the two estimation. The design of the loss function is rooted in estimating a good importance weight, and then use the importance weight to align the latent feature space. However, unlike a simple competing paradigm like min max optimization, in there, the importance weighting estimation scheme only makes sense when HFA is achieved. And we can only get closer to HFA (almost impossible to achieve perfect HFA as shown in equation (2)) with a good importance weight estimation. This is especially concerning because the estimation bound can be arbitrarily large, which makes it looks difficult from an optimization perspective.

Overall, it feels like there is a hole that needs to be filled between the entangled importance weight and HFA condition, that is, what happens when HFA is not perfectly achieved (which is almost surely the case), should we change the w estimation strategy and estimation characterization?

**Essential References Not Discussed:**

Not to my knowledge.

**Experimental Designs Or Analyses:**

Yes.

**Methods And Evaluation Criteria:**

Yes.

**Other Comments Or Suggestions:**

Minor Comments:

There are confusing typos throughout the paper. For example, page3 line 116 second column, it should be under HFA assumption correct?

**Other Strengths And Weaknesses:**

See above.

**Questions For Authors:**

Questions:

1. Regarding Theorem 3.5, the result seems rather counter intuitive, because the total variation distance between the two measures can be arbitrarily large when the probability of one class diminishes in distribution shift ($\rho \rightarrow 0$). This kind of implies the bound could be potentially vacuous. Can the authors clarify on this issue?
2. The same concern goes with Theorem 3.8, as it might be the case that for a "bad" predictor, the importance weight could also be arbitrarily large ($\sigma_{min}(\mathbf{C}) \righarrow 0$). Is there any guarantee or general explanation on how can this be avoided?

If these two answers and the previous concern is addressed, I would be happy to raise my score, as I do find the theoretical framework interesting.

**Relation To Broader Scientific Literature:**

The paper provides an interesting piece of theoretical analysis for domain/distribution shift.

**Theoretical Claims:**

Yes.

---

> ### Author Rebuttal · Authors · 2025-03-31
>
> Q1. Overall, it feels like there is a hole that needs to be filled between the entangled importance weight and HFA condition, that is, what happens when HFA is not perfectly achieved (which is almost surely the case), should we change the $\mathbf w$ estimation strategy and estimation characterization?
>
> A1. We appreciate your insightful observation. In practice, achieving perfect Heterogeneous Feature Alignment (HFA) is challenging, which may introduce biases in importance weight estimation. To mitigate this, we adopt a joint optimization approach that simultaneously optimizes Importance Weight Estimation and Heterogeneous Feature Alignment, allowing the two modules to interact synergistically and form a positive feedback loop that enhances overall performance. Specifically, better HFA reduces distribution mismatch, leading to more accurate importance weight estimation, while improved weight estimation provides a more reliable training signal, further refining feature alignment. Of course, when there is a significant deviation in HFA, the estimated importance weight $\mathbf{w}$ may become biased due to residual feature mismatches between domains, which will further affect the adaptation performance. In this case, we need to adjust the $\mathbf w$ estimation strategy. For example, prior information can be incorporated to design regularization terms that correct $\mathbf w$.
>
> Q2. Regarding Theorem 3.5, the result seems rather counter intuitive, because the total variation distance between the two measures can be arbitrarily large when the probability of one class diminishes in distribution shift ($\rho \rightarrow 0$). This kind of implies the bound could be potentially vacuous. Can the authors clarify on this issue?
>
> A2. Thanks for your insightful comments. Theorem 3.5 gives an upper bound on the discrepancy of the feature distribution conditional on the label. We think it is not counter-intuitive since $\rho \rightarrow 0$ only implies a loose upper bound and does not imply that the arbitrarily large total variation. In other words, an infinite upper bound is not equivalent to an infinite total variation. In fact, as shown in Eq. (1), the total variation is a value within the range [0,1]. In addition, Theorem 3.5 reflects the natural limitation that, in extreme cases of label shift, adaptation becomes fundamentally more difficult. Most theoretical results in domain adaptation and importance weighting [1, 2] similarly exhibit such behavior as $\rho \rightarrow 0$. In this paper, we mainly consider the mild case consistent with the mainstream Label Shift [2, 3], and first give a theoretical analysis of the brand-new HLS problem. Based on the theory, we propose a bound minimization HLS framework and achieve good performance. In fact, the extreme imbalance problem has always been an important puzzle, and how to construct more tight upper bounds in this case is a very interesting research problem.
> $$d_{TV}(P,Q)= \mathop {\sup }\limits_{E~\text{is measurable}}\left|P(X\in E)-{Q} (X\in E)\right|.~~~~(1)$$
>
> [1] Remi Tachet des Combes, Han Zhao, Yu-Xiang Wang, Geoffrey J. Gordon: Domain Adaptation with Conditional Distribution Matching and Generalized Label Shift. NeurIPS 2020.
>
> [2] Kamyar Azizzadenesheli, Anqi Liu, Fanny Yang, Animashree Anandkumar: Regularized Learning for Domain Adaptation under Label Shifts. ICLR (Poster) 2019.
>
> [3] Ruidong Fan, Xiao Ouyang, Tingjin Luo, Dewen Hu, Chenping Hou: Incomplete Multi-View Learning Under Label Shift. IEEE Trans. Image Process. 32: 3702-3716 (2023).
>
> Q3. The same concern goes with Theorem 3.8, as it might be the case that for a "bad" predictor, the importance weight could also be arbitrarily large (${\sigma _{\min }(\mathbf{C})}\rightarrow 0$). Is there any guarantee or general explanation on how can this be avoided?
>
> A3. Thanks for your insightful comments. Theorem 3.8 gives an upper bound for Importance Weight Estimation. ${\sigma _{\min }(\mathbf{C})}\rightarrow 0$ leads to a loose upper bound for weight estimation error. Similarly, an infinite upper bound is not equivalent to an arbitrarily large weight estimation error. It typically occurs when the model is poorly trained or when the source and target domains have extreme shifts. In our paper, to avoid this problem, as mentioned in Appendix B.2, we first warm up HLSAN by training without weighting for the first 20 epochs to obtain an acceptable predictor, and then we build estimators for $\mathbf{C}$. In this case, $\mathbf{C}$ is a diagonally dominant real symmetric probability matrix, which guarantees that $\mathbf{C}$ is invertible and its singular values are all greater than 0. Through such way, the extreme cases suggested by the reviewer are largely avoided. Since this issue is a common challenge for label shift, and we have employed certain strategies to mitigate it as much as possible, how to fully resolve it from a theoretical perspective remains an open problem.

---

> > ### Comment · Reviewer_kepV · 2025-04-02
> >
> > Thank you for the answers. Here are my follow up responses.
> >
> > A1: I think the authors response somewhat confirmed my question in the sense that there is no theoretical characterization of estimation strategy for the imperfect case, and that is totally fine with empirical evaluation.
> >
> > A2: Again the answer confirmed my understanding that the current bound is rather vacuous in the sense that it is generally much larger than the actual error. Even though this hinders the value of the theoretical results from my perspective, but it is not a deal breaker considering the state of current literature as provided by the authors.
> >
> > A3: An empirical way of avoiding the problem sounds ok, and thank you for clarifying the question.
> >
> > Finally, I would like to thank the authors for their response, and my core concerns have mostly been empirically addressed, so I would like to raise my score to 3.

---

### Official Review · Reviewer_cVaJ · 2025-03-11

**Overall Recommendation:** 3

**Summary:**

The paper introduces the concept of Heterogeneous Label Shift to address cross-modal knowledge transfer challenges, where both feature heterogeneity and shifted label distributions affect model performance. It presents an error decomposition theorem and a bound minimization framework that tackle these issues. Extensive experiments validate the effectiveness of this approach.

**Claims And Evidence:**

The claims made in the submission are supported by evidence.

**Essential References Not Discussed:**

No, essential prior works are appropriately cited and discussed.

**Experimental Designs Or Analyses:**

I have generally checked the experimental design and analysis, and they appear sound.

**Methods And Evaluation Criteria:**

Yes, the proposed methods and the constructed benchmark make sense for the problem at hand.

**Other Comments Or Suggestions:**

I have no other suggestions.

**Other Strengths And Weaknesses:**

No additional strengths or weaknesses are observed beyond those already mentioned.

**Questions For Authors:**

1. What is the primary role of the unlabeled parallel instances, and would the proposed method become inapplicable if these data were lacking?
2. If the unlabeled parallel instances are crucial, should the experiments analyze the impact of the number of these instances on the final performance?
3. While there is existing work on both heterogeneous domain adaptation and label shift, what unique challenges does the Heterogeneous Label Shift problem introduce?

**Relation To Broader Scientific Literature:**

The paper contributes to the study of heterogeneous domain shift by focusing on heterogeneous label shift.

**Theoretical Claims:**

I have generally checked the proofs, but some details have not been thoroughly verified.

---

> ### Author Rebuttal · Authors · 2025-03-31
>
> Q1. What is the primary role of the unlabeled parallel instances, and would the proposed method become inapplicable if these data were lacking?
>
> A1. Thank you for your insightful question. As the second response to the Reviewer SzhU, parallel instances establish a cross-modal channel that facilitates effective knowledge transfer, linking the source and target domains while mitigating extreme negative transfer, as discussed in Appendix B.3. Thus, our method has a reliance on parallel instances. In reality, parallel instances are common and easy to obtain compared to labeled target data. We then investigate this problem setting and propose an theoretical inspired models. Besides, we also recognize the challenges due to the lack of parallel instances. In future work, we will continue to explore strategies such as self-supervised alignment, semantic similarity, and contrastive learning to explore the task of cross-modal knowledge transfer without parallel instance.
>
> Q2. If the unlabeled parallel instances are crucial, should the experiments analyze the impact of the number of these instances on the final performance?
>
> A2. Thank you for the insightful suggestion. To validate this, we have conducted preliminary experiments to analyze the impact of the number of parallel instances on final performance and the results are shown in the following tables. More thorough investigations will be added to the final version. Based on these observations, we can see that performance gradually improves as more parallel instances are available.
> # The influence of Parallel Instances for HLSAN.
> |Task| 50 | 100 | 150 |200|
> |-----------------|-------------------|--------------------|--------------------|--------------------|
> | SP$\rightarrow$EN |49.2|53.5|54.3|56.2|
> | SP$\rightarrow$FR |52.1|54.7|56.1|56.8|
> | SP$\rightarrow$GE|48.1|50.2|52.1|55.8|
> | SP$\rightarrow$IT|44.3|46.4|50.1|53.2|
> | Wiki T$\rightarrow$I|79.7|81.7|84.6|87.3|
>
> Q3. While there is existing work on both heterogeneous domain adaptation and label shift, what unique challenges does the Heterogeneous Label Shift problem introduce?
>
> A3. Thanks for your question. We think coupled Heterogeneous Feature and Label Shift is the unique challenges of HLS problem. Unlike traditional label shift, where only the label distribution changes, HLS involves simultaneous shifts in both feature space and label distribution, making standard importance-weighting methods insufficient. Unlike traditional heterogeneous domain adaptation, HLS considers the more general case where heterogeneous features are accompanied by label shift, which makes the existing feature-alignment methods fail to achieve joint distribution alignment. More seriously, our HLS problem can not be decoupled into two sequent or parallel problems since Heterogeneous Feature Alignment and Importance Weight Estimation are intertwined. It is not ideal to simply use existing heterogeneous domain adaptation and label shift techniques to solve the two problems separately. To solve this problem, HLSAN designs two interactive modules of Heterogeneous Feature Alignment and Importance Weight Estimation respectively. By interacting them synergistically, we form a positive feedback loop that enhances overall performance.

---

### Official Review · Reviewer_Aznb · 2025-03-14

**Overall Recommendation:** 3

**Summary:**

The paper introduces a new Heterogeneous Label Shift (HLS) method to tackle heterogeneous label shift. After analyzing the impact of feature spaces and label shift, the authors propose a new error bound. They show, with experiments on 2 real-life datasets, the efficiency of their method for multi-modal data.

**Claims And Evidence:**

The authors claim that in heterogeneous scenarios, "label shift is inevitable." In my opinion, this is not so obvious, and I would like to have some references to show the necessity of this application. Especially when, in the experiments, the label shift is made artificially.

**Essential References Not Discussed:**

NA

**Experimental Designs Or Analyses:**

My only concern is the ablation study that is done only over 2 tasks, while other experiments are done over 13 tasks. cf questions.

**Methods And Evaluation Criteria:**

The dataset makes sense for the heterogeneous Domain Adaptation, but maybe not for label shift without making them shift.

**Other Comments Or Suggestions:**

- typo Table 1: discrete sourece -> discrete source

- Figure 2 is complex to understand. It could be nice to have meaningful colors to explain what is learnable and fixed.
I don't understand why a network like encoders represents the latent space. I don't see the shared network encoder in the paper. Maybe put the name of each to understand better the illustration (i.e., Ts, Tt, h ...).

- You use T to name the encoder, and in eq. 12 and 13, you are using f. Is it the same or a different encoder?

- It can be nice to have a small summary of the compared methods.

- I suggest putting the related work part at the beginning. At this point of the paper, we want to continue reading your method.

**Other Strengths And Weaknesses:**

Strengths:
- The paper deals with an interesting problem in domain adaptation that has not been treated much.

Weaknesses:
- the paper is very dense and could be hard to follow.
- The notation seems to change in the paper. cf comments

**Questions For Authors:**

- I don't get why a heterogeneous domain should imply a label shift. You can distribute the label in two different modalities. Even in the experiment, the label shifts are simulated.

- The ablation study is done only over two adaptations SP -> EN and SP -> IT. Can we have the ablation study for both experiments' overall adaptations? You choose to take the worst case $\gamma = 10$; what are the results for smaller $\gamma$?

**Relation To Broader Scientific Literature:**

NA

**Theoretical Claims:**

I checked briefly the proofs that seemed good, but I did not have time to go in depth.

---

> ### Author Rebuttal · Authors · 2025-03-31
>
> Q1. Typos and suggestions.
>
> A1. Thank you for your valuable feedback. We have corrected the typo in the manuscript and checked it carefully throughout. In addition, as suggested by the reviewers, we added a brief summary of the comparison methods, and the related work section has been moved earlier to improve the logical flow. Finally, we have revised Figure 2 with meaningful colors to distinguish learnable and fixed components, added explicit labels (e.g., $T_s, T_t, h$), and clarified the role of the encoder.
>
> Q2. I don't get why a heterogeneous domain should imply a label shift. You can distribute the label in two different modalities. Even in the experiment, the label shifts are simulated.
>
> A2. Thank you for your comment. In reality, since the collected data come from different modalities, it is more likely to cause label shift. For example, when images and text data are collected from different websites, it is almost impossible to maintain the same category ratio of picture and text data, which unavoidably cause the label shift problem. Of course, as the reviewer said, we can distribute label in two different modalities to avoid label shift. Nevertheless, in practice, this approach is not feasible since there is no label data in target domain. In fact, label shift is a well-documented phenomenon in domain adaptation and transfer learning [1,2], and the HLS problem is also raised from reality. Regarding the experiments, in order to evaluate our approach from multiple angles and multiple label shift cases, we follow standard setup in label shift research [3,4] by simulating label shifts to systematically evaluate our method. This allows us to isolate the impact of label shift and better analyze its effects.
>
> [1] Zachary C. Lipton, Yu-Xiang Wang, Alexander J. Smola: Detecting and Correcting for Label Shift with Black Box Predictors. ICML 2018: 3128-3136.
>
> [2] 	Ruihan Wu, Chuan Guo, Yi Su, Kilian Q. Weinberger: Online Adaptation to Label Distribution Shift. NeurIPS 2021: 11340-11351.
>
> [3] Kamyar Azizzadenesheli, Anqi Liu, Fanny Yang, Animashree Anandkumar: Regularized Learning for Domain Adaptation under Label Shifts. ICLR (Poster) 2019.
>
> [4] Ruidong Fan, Xiao Ouyang, Tingjin Luo, Dewen Hu, Chenping Hou: Incomplete Multi-View Learning Under Label Shift. IEEE Trans. Image Process. 32: 3702-3716 (2023).
>
> Q3. The ablation study is done only over two adaptations SP$\rightarrow$EN and SP$\rightarrow$IT. Can we have the ablation study for both experiments' overall adaptations? You choose to take the worst case $\gamma=10$; what are the results for smaller $\gamma$?
>
> A3. Thank you for your suggestions. Due to space constraints, we conducted the ablation study on two representative adaptations (SP$\rightarrow$EN and SP$\rightarrow$IT) and selected $\gamma=10$ as Illustration. In fact, we have done all the experiments. As shown in the following tables. The results can also reveal the same two key findings. 1) Removing any component degrades performance, demonstrating the importance of each term. 2) Incorporating with importance weight enhances performance, indicating the necessity of aligning label distribution shifts.
>
> # The accuracy (%) of ablation study of HLSAN with $\gamma = 10$.
> |Task| w/o/D | w/o/P | w/o/W |HLSAN|
> |-----------------|-------------------|--------------------|--------------------|--------------------|
> | SP$\rightarrow$EN |50.1|49.6|54.7|**55.9**|
> | SP$\rightarrow$FR |53.7|51.7|55.4|**57.3**|
> | SP$\rightarrow$GE|53.4|49.4|54.0|**56.3**|
> | SP$\rightarrow$IT|44.4|42.5|48.3|**55.1**|
> | Wiki T$\rightarrow$I|75.9|74.6|77.2|**85.8**|
>
> # The accuracy (%) of ablation study of HLSAN with $\gamma = 5$.
> |Task| w/o/D | w/o/P | w/o/W |HLSAN|
> |-----------------|-------------------|--------------------|--------------------|--------------------|
> | SP$\rightarrow$EN |49.1|47.3|53.1|**57.4**|
> | SP$\rightarrow$FR |48.8|48.6|50.2|**53.7**|
> | SP$\rightarrow$GE|47.7|46.5|51.6|**53.5**|
> | SP$\rightarrow$IT|46.9|45.5|48.2|**50.8**|
> | Wiki T$\rightarrow$I|73.3|72.6|75.2|**77.6**|
>
> # The accuracy (%) of ablation study of HLSAN with $\gamma = 2$.
> |Task| w/o/D | w/o/P | w/o/W |HLSAN|
> |-----------------|-------------------|--------------------|--------------------|--------------------|
> | SP$\rightarrow$EN |48.2|47.3|52.1|**53.5**|
> | SP$\rightarrow$FR |50.1|48.6|52.4|**54.7**|
> | SP$\rightarrow$GE|46.8|45.4|47.9|**50.2**|
> | SP$\rightarrow$IT|41.3|40.5|44.2|**46.4**|
> | Wiki T$\rightarrow$I|72.9|70.6|76.2|**81.7**|

---

> > ### Comment · Reviewer_Aznb · 2025-04-04
> >
> > I thank the reviewers for their answers. All my questions have been answered. Their work is valuable and can interest the community, so I increased my score.

---

### Official Review · Reviewer_SZhU · 2025-03-17

**Overall Recommendation:** 3

**Summary:**

This paper introduces Heterogeneous Label Shift (HLS), a problem where cross-modal knowledge transfer must address simultaneous heterogeneous feature spaces and shifted label distributions. The work presents a theoretical error decomposition, proposes a bound minimization framework (HLSAN), and validates it empirically on cross-modal classification tasks. Key contributions include formalizing HLS, deriving theoretical guarantees, and demonstrating superior performance over baseline methods.

**Claims And Evidence:**

Yes

**Essential References Not Discussed:**

Cross-Modal DA: CLIP (Radford et al., 2021) or ViLBERT (Lu et al., 2019) for vision-language alignment.
Unsupervised Weight Estimation: Kernel Mean Matching (Huang et al., 2007) or IWCV (Sugiyama et al., 2007).

**Experimental Designs Or Analyses:**

Text→Text: Spanish→English/French/German/Italian with TF-IDF features (Reuters).
Image→Text: Wikipedia articles with BiT-M (image) and BigBird (text) features.

**Methods And Evaluation Criteria:**

Yes

**Other Comments Or Suggestions:**

Explore self-supervised alignment (e.g., contrastive learning) to reduce dependence on parallel data.
Compare to transformer-based models (e.g., CLIP) for cross-modal transfer.
Analyze scalability to large-scale datasets (e.g., ImageNet-21K).

**Other Strengths And Weaknesses:**

Strengths:
1.Novel problem formulation with practical relevance to cross-modal applications.
2.Theoretically grounded framework integrating feature alignment and label shift adaptation.
Weaknesses:
1.Reliance on parallel instances (O) for initialization may limit real-world applicability.
2.Computational cost of alternating optimization (L_O and L_KT) is not analyzed.

**Questions For Authors:**

1.How does HLSAN scale with the number of classes (k)? Does the RQP estimator degrade for k > 100k>100?
2.Can the framework handle non-parallel modalities (e.g., audio→text) without instance pairs O?
3.What is the computational overhead of adversarial alignment vs. standard DANN?

**Relation To Broader Scientific Literature:**

Label Shift: Extends Lipton et al. (2018b)’s confusion matrix approach to heterogeneous settings via RQP.
Heterogeneous DA: Builds on Fang et al. (2023)’s semi-supervised HDA but addresses label shift via adversarial alignment.
Adversarial DA: Connects to DANN (Ganin et al., 2016) but aligns reweighted source/target distributions.

**Theoretical Claims:**

Error Decomposition Theorem (Theorem 3.3)
Assumes invertible confusion matrix (Lemma 3.7), which may fail in high-dimensional settings.

---

> ### Author Rebuttal · Authors · 2025-03-31
>
> Q1. How does HLSAN scale with the number of classes ($k$)? Does the RQP estimator degrade for $k > 100$?
>
> A1. Thank you for your thoughtful question. Based on our theoretical analysis (Theorem 3.8) as shown in the following formula, the weight estimation error is sublinear with respect to $k$ so HLSAN can efficiently scale with $k$. Of course, when k is large, such as $k \leq100$, it will inevitably reduce the performance of the RQP estimator due to the increased classification difficulty. In fact, weight estimation with large number of classes has been a recognized difficult problem in label shift. But by increasing the sample size we can mitigate that the impact of the increase in categories at this time.
> $$||\hat{\mathbf{w}} - \mathbf{w}||_2\le c_1\sqrt {\frac{{72k}}{{{n_s}}}\log \left( {\frac{{12k}}{\delta }} \right)}+ c_2\sqrt {\frac{{9k}}{{{n_t}}}\log \left( {\frac{{6k}}{\delta }} \right)}.$$
>
> Q2. Can the framework handle non-parallel modalities (e.g., audio→text) without instance pairs $\mathcal O$?
>
> A2. Thank you for your insightful question. Parallel instances provide a cross-modal channel, which enables HLSAN to perform effective knowledge transfer. Different from previous settings with some assumptions to connect source and target domains, instance pairs are vital for our approach since there is no labeled data in the target domain. Therefore, it is difficult for HLSAN to handle non-parallel modalities without instance pairs $\mathcal O$. In fact, we investigate this setting since the parallel instances are now common and easily accessible, such as Image-text data, news broadcasts. We recognize the challenges due to the lack of parallel instances. In the future, we'll explore strategies such as self-supervised alignment, semantic similarity, and contrastive learning to enhance performance in this case.
>
> Q3. What is the computational overhead of adversarial alignment vs. standard DANN?
>
> A3. We appreciate the reviewer’s concern regarding the computational cost. Compared to standard DANN, adversarial alignment introduces additional computations primarily due to the alternating optimization introduces additional updates for $L_O$ and $L_{KT}$. Specifically, each training step requires solving two subproblems sequentially, which adds approximately 60% increase in training time per epoch. Fortunately, our method can level off within 30 epochs in most cases, and the additional computational cost is also affordable in real applications.

---

### Decision · Program_Chairs · 2025-05-01

**Decision:**

Accept (poster)

**Comment:**

This paper introduces Heterogeneous Label Shift (HLS), a new learning problem addressing knowledge transfer when both feature spaces and label distributions differ between source and target domains. It provides a theoretical error decomposition and proposes the HLSAN algorithm, demonstrating its effectiveness in cross-modal classification tasks with varying label shifts.

The main strengths of this paper are:
1. HLS is a new and exciting framework that generalizes label shift in a way that encompasses more applications.
2. The theoretical results not only show that their algorithm achieves theoretically good performance, but also provides some fundamental facts about the problem at hand.

Addressing concerns from reviewers:
1. Two assumptions were brought up by the reviewers.  The existence of parallel instances and the invertibility of the confusions matrix.
2. What happens if HFA does not precisely hold, and how does this interact with the weighting scheme?
3. Providing more ablation studies

Regarding (1), I think that without these assumptions we lose identifiability and the problem becomes impossible (from what I can tell).  (2) is outside of the scope of this work, and typically, model misspecification is handled in followup works.  (3) was addressed by the authors.